# EMA Without the Lag: Bias-Corrected Iterate Averaging Schemes

## Abstract

Stochasticity in language model fine-tuning, often caused by the small batch sizes typically used in this regime, can destabilize training by introducing large oscillations in generation quality. A popular approach to mitigating this instability is to take an Exponential moving average (EMA) of weights throughout training. While EMA reduces stochasticity, thereby smoothing training, the introduction of bias from old iterates often creates a lag in optimization relative to vanilla training. In this work, we propose the Bias-Corrected Exponential Moving Average (BEMA), a simple and practical augmentation of EMA that retains variance-reduction benefits while eliminating bias. BEMA is motivated by a simple theoretical model wherein we demonstrate provable acceleration of BEMA over both a standard EMA and vanilla training. Through an extensive suite of experiments on Language Models, we show that BEMA leads to significantly improved convergence rates and final performance over both EMA and vanilla training in a variety of standard LM benchmarks, making BEMA a practical and theoretically motivated intervention for more stable and efficient fine-tuning.

## 1 Introduction

With the increasing scale of Language Models (LMs), serious limitations on the quantity of new, high-quality data available for pre- and post-training have led to a renewed interest in understanding optimization and how best to use scarce data (Villalobos et al., 2024; Muennighoff et al., 2023). Indeed, in regimes where the number of distinct, high-quality sequences of text is limited, e.g. in finetuning on corpora where data collection is expensive such as math (Liu et al., 2023; Hendrycks et al., 2021), code (Austin et al., 2021), or specialized domain expertise (Lee et al., 2020; Chalkidis et al., 2020), practitioners are often forced to use a small batch size in order to squeeze as much information out of the data as possible (Masters & Luschi, 2018; Rather et al., 2024; Zhang et al., 2024). While small batch sizes allow for more gradient steps to be taken, they come at the cost of increased variance in the stochastic gradients, which can lead to instability.

Training instability is particularly pronounced in situations where a model is evaluated *closed loop*, i.e. a model is rolled out by iterative application on its own outputs; such closed-loop rollout effects were originally observed in the context of imitation learning (Ross & Bagnell, 2010; Ross et al., 2011; Chang et al., 2023; Block et al., 2024) and are a result of small learning errors in each step of a rollout being catastrophically amplified through repeated application. LMs exhibit the same pathology because errors occurring at the *token* level are repeatedly fed back into the model due to the autoregressive nature of generation; this connection between imitation learning and LMs has been explored extensively in the literature (Chang et al., 2023; Block et al., 2023; 2024; Foster et al., 2024; Rohatgi et al., 2025). In Block et al. (2024), the authors observed that error amplification in the context of closed-loop evaluation often results from stochasticity in the gradients, which they term *Gradient Variance Amplification* (GVA), and can substantially degrade model performance even when cross-entropy loss is small; due to the many downstream problems that GVA can cause, Block et al. (2024) recommends focusing on designing *stabilizers* that mitigate these effects.

The most empirically successful approach to stabilization is *iterate averaging*, wherein the training trajectory is postprocessed by applying a weighted average to the individual iterates in order to reduce variance, with the most popular such averaging scheme being an *Exponential Moving Average* (EMA) (Ruppert, 1988; Polyak & Juditsky, 1992; Izmailov et al., 2018; Sandler et al., 2023;

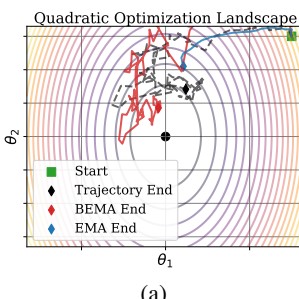 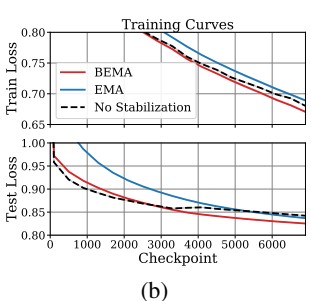 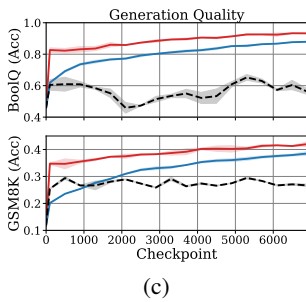

(a)           (b)           (c)

Figure 1: **(a)** Example trajectory of stochastic quadratic optimization in two dimensions, stabilized both by EMA and BEMA; the vanilla trajectory does not converge to the minimum due to gradient variance, while EMA induces significant slowing down due to bias; only BEMA converges to the minimum quickly. **(b)** Train and test loss curves and **(c)** BoolQ and GSM8K benchmarks, for Qwen2.5-1.5B, without stabilization and with both EMA and BEMA. While EMA and BEMA improve performance over vanilla training, BEMA achieves better accuracy more quickly than EMA.

Busbridge et al., 2023). In deep learning, EMA has seen great success both in stabilizing training (Block et al., 2024) and in improving the final performance of the model (Izmailov et al., 2018), but the variance reduction comes at the cost of introducing bias from earlier iterates, which empirically manifests as a *lag* in the training trajectory: while the training curves of EMA are typically signficantly smoother than the optimization with no stabilization, they often converge more slowly. This observation naturally leads to the following question: *Can we design a stabilizer achieving the benefits of EMA without the lag?*

We answer in the affirmative by introducing a new stabilizer, BEMA (summarized in Algorithm 1), which achieves the best of both worlds. We adopt a theoretical model (Sections 2 and 3) inspired and justified by prior empirical work in deep learning and optimization (Duchi et al., 2011; Gupta et al., 2018; Zhang et al., 2019; Vyas et al., 2024) and derive BEMA as the optimal stabilizer in this model. We then discuss the practical implementation of BEMA in Section 4, and **observe that it is a drop-in replacement for the commonly used EMA stabilizer, requiring changing only two lines of code**. Finally, we evaluate BEMA on a variety of Language Model (LM) finetuning tasks in Section 5, where we find that **BEMA significantly outperforms both vanilla training and EMA across a wide range of tasks**. A survey of related work can be found in Appendix B, further empirical results can be found in Appendices A and D, and all proofs are deferred to Appendix E.

## 2 MATHEMATICAL PRELIMINARIES

We are interested in the problem of stabilizing the training of language models (LMs) when the optimizer has a sufficiently small batch size so as to make gradient stochasticity a significant problem for closed-loop evaluation.[1] We formalize a language model as a conditional distribution $p_\theta(y|x)$ parameterized by some weight $\theta$, where $y \in \mathcal{V}$ is the next *token*, which is a member of the vocabulary $\mathcal{V}$ and $x \in \mathcal{V}^*$ is a *prompt* or *context* consisting of a sequence of tokens. In this paper, we are primarily interested in *Supervised Fine Tuning* (SFT), wherein we are given a dataset $\mathcal{D}$ consisting of sequences and we attempt to maximize the log likelihood of a given sequence, i.e., minimize $-\mathbb{E}_{(x,y)\sim\mathcal{D}}\left[\log p_\theta(y|x)\right]$. Due to the high dimension of the weights $\theta$, this optimization is typically accomplished via stochastic local search techniques. Because we are interested in a model's performance on closed-loop rollouts (via autoregressive generation), the oscillations in model performance throughout training observed in Block et al. (2024) pose a significant problem, which motivates the need for stabilizing the optimization process, which we now discuss.

In order to formalize the notion of a *stabilizer*, we consider the classical setting of stochastic optimization (Robbins & Monro, 1951; Ruppert, 1988; Polyak & Juditsky, 1992; Nesterov, 2013), where we are given a function $f : \mathbb{R}^d \to \mathbb{R}$ taking its minimum at 0 and access to a stochastic

---

[1]While the primary focus is LMs, we conjecture that our approach can be applied to other situations in which GVA presents a problem, such as in Imitation Learning (Block et al., 2024).

gradient oracle that returns a noisy gradient $\nabla f(\theta - \mu^\star) + \xi$ when queried at a point $\theta \in \mathbb{R}^d$ for fixed minimum $\mu^\star \in \mathbb{R}^d$, where $\xi$ is some random vector representing the noise. We focus on the simplest algorithm for this problem, stochastic gradient descent (SGD), which updates the parameter $\theta$ according to the rule $\theta_{t+1} = \theta_t - \eta_t \left( \nabla f(\theta_t - \mu^\star) + \xi_t \right)$.

To aid analytical tractability, we will follow Mandt et al. (2015); Li et al. (2017); Malladi et al. (2022) and consider the continuous time limit of this process assuming that $\xi$ is mean zero and has finite second moment, which is given by the stochastic differential equation (SDE):

$$d\theta_t = -\nabla f(\theta_t - \mu^\star) \, dt + \sqrt{\eta} \cdot \boldsymbol{\Sigma} \, dW_t, \qquad \theta_0 \in \mathbb{R}^d, \tag{1}$$

where $\Sigma \in \mathbb{R}^{d \times d}$ can be determined by the covariance of the noise in the stochastic gradient oracle, $\eta$ is the (constant) scale on the learning rate, and $W_t$ is a standard Brownian motion in $\mathbb{R}^d$.[2] We will adopt the perspective of stochastic optimization as a statistical parameter estimation problem, where the minimizer we seek is the parameter we wish to estimate (Robbins & Monro, 1951; Polyak & Juditsky, 1992; Ruppert, 1988; Nesterov, 2013) and suppose that the stabilizer is some algorithm that is given an optimization trajectory and aims to return an estimate of the minimum. More formally, our goal is the following.

> *For a fixed, finite horizon $T > 0$, given access to the trajectory $(\theta_t)_{0 \leq t \leq T}$,*
> *how can we best estimate $\mu^\star$ in a memory and computationally efficient way?*

The ultimate goal is to construct an algorithm that improves optimization in practice, particularly in the context of finetuning language models; due to the size of LMs, the memory efficiency of the final estimator is crucial. While LMs themselves are certainly not convex functions with respect to their parameters $\theta$, recent work has demonstrated that they can be locally well-approximated by a quadratic function and optimization insights arising from this regime often carry over to the more practical (and less analytically tractable) non-convex case of modern-day transformers, especially in the finetuning regime (Jacot et al., 2018; Gupta et al., 2018; Cohen et al., 2021; Malladi et al., 2023; Vyas et al., 2024). Thus, to further simplify our problem and permit us to develop a concrete, practical algorithm, we will focus our theory on the noisy quadratic model (Zhang et al., 2019), where $f(\theta) = \frac{1}{2}\theta^\top \mathbf{A}\theta$ for some positive definite matrix $\mathbf{A} \in \mathbb{R}^{d \times d}$, which can be interpreted as the Hessian of the loss of the LM at some fixed $\theta_0$; to reiterate, while this assumption does not hold for real LMs, it provides a useful testbed from which to develop intuition and algorithmic interventions (Duchi et al., 2011; Gupta et al., 2018; Vyas et al., 2024).

It has long been known that the continuous time limit of SGD applied to a quadratic loss is the Ornstein-Uhlenbeck (OU) process (Mandt et al., 2015), where (1) becomes:

$$d\theta_t = \mathbf{A}(\mu^\star - \theta_t) \, dt + \sqrt{\eta} \cdot \boldsymbol{\Sigma} \, dW_t, \qquad \theta_0 \in \mathbb{R}^d, \tag{2}$$

where we assume that $\mathbf{A}, \boldsymbol{\Sigma} \in \mathbb{R}^d$ are symmetric positive definite matrices. It is standard that the OU process admits a simple closed form, given in Appendix E, that we use extensively in our analysis.

## 3 Optimal Stabilization in Stochastic Quadratic Optimization

Above, we formalized stabilization as the statistical estimation problem of estimating $\mu^\star$ given access to a single trajectory $(\theta_t)_{0 \leq t \leq T}$ of the OU process defined in (2). For the sake of simplicity, we will in this section assume that $\boldsymbol{\Sigma} = \sigma^2 \mathbf{I}$ and defer the general case (as well as all proofs) to Appendix E. Before we proceed, we first present a standard lower bound on the expected squared error of any estimator $\widehat{\mu}^\star$ using the Cramer-Rao and van Trees inequalities (Lehmann & Casella, 2006); asymptotic versions of this standard bound can be found in Liptser & Shiryaev (2013a;b); Kutoyants (2013).

**Proposition 1.** *For any fixed $T < \infty$, let $(\theta_t)_{0 \leq t \leq T}$ be a trajectory from (11) with $\boldsymbol{\Sigma} = \sigma^2 \mathbf{I}$ and suppose that $\widehat{\mu}$ is an estimator of $\mu^\star$ measurable with respect to the filtration generated by $(\theta_t)_{0 \leq t \leq T}$. Suppose further that $\widehat{\mu}$ is unbiased, i.e. $\mathbb{E}[\widehat{\mu}] = \mu^\star$. Then it holds that*

$$\mathbb{E}\left[\|\widehat{\mu} - \mu^\star\|^2\right] \geq \frac{\eta \sigma^2 \cdot \operatorname{Tr}\left(\mathbf{A}^{-2}\right)}{T}. \tag{3}$$

---

[2]We will always assume that at least a weak solution to (1) exists and is unique, which is certainly the case for the OU process on which we focus. For more details on SDEs, see Le Gall (2016).

*More generally, if the bias of $\widehat{\mu}$ is a contraction, i.e., the map $\mu^\star \mapsto \mathbb{E}_{\mu^\star}[\widehat{\mu} - \mu^\star]$ is L-Lipschitz for some $L < 1$, then (3) holds with a prefactor of $(1 - L)^2$.*

While the asymptotic performance (as $T \uparrow \infty$) of a number of standard estimators is well understood (Kutoyants, 2013), in this work we are interested in what occurs for finite $T$, which is the regime of interest in practice. Perhaps the simplest approach to estimating $\mu^\star$ is that adopted by vanilla optimization: simply take the final iterate $\theta_T$ as the desired estimate. In this case, we can precisely compute the expected squared error of this estimate, which is given in the following proposition.

**Proposition 2.** *Let $(\theta_t)_{0 \le t \le T}$ be a trajectory from (11) with $\Sigma = \sigma^2 \mathbf{I}$. Then it holds that*

$$\mathbb{E}\left[\|\theta_T - \mu^\star\|^2\right] = \left\|e^{-\mathbf{A}T}(\mu^\star - \theta_0)\right\|^2 + \eta\sigma^2 \cdot \mathrm{Tr}\left(\mathbf{A}^{-1}\left(\mathbf{I} - e^{-2\mathbf{A}T}\right)\right) \tag{4}$$

While the last iterate estimator is attractive in its simplicity, and the first term in (4) decays exponentially quickly, it leaves much to be desired because it is not consistent, i.e., $\theta_T \not\to \mu^\star$ even as $T \uparrow \infty$, unless $\eta \downarrow 0$. This is a simple example of the well-understood phenomenon in stochastic optimization that motivates learning rate decay. Absent computational constraints, it may well be advisable to train at a small (or aggressively decayed) learning rate; unfortunately, the number of optimizer steps required to reach time $T$ in the discrete approximation scales as $T/\eta$, which quickly becomes prohibitive for small $\eta$. Thus, practitioners often wish to train at as high a learning rate as possible in order to accelerate convergence (Smith & Topin, 2019; Loshchilov & Hutter, 2022), which helps explain why some degree of trajectory stabilization has become commonplace.

The most common approach to stabilizing training in modern deep learning, beyond learning rate decay, is to apply iterate averaging (Izmailov et al., 2018; Block et al., 2024; Busbridge et al., 2023), specifically an exponential moving average (EMA) of the model parameters (Ruppert, 1988; Polyak & Juditsky, 1992), which is defined as $\widehat{\mu}_t^{\mathsf{EMA}} = (1 - \alpha_t)\widehat{\mu}_t^{\mathsf{EMA}} + \alpha_t \theta_t$ with $\theta_t^{\mathsf{EMA}} = \theta_0$ for some sequence of weights $\alpha_t \in (0, 1)$. While different choices for $\alpha_t$ are possible and discussed in the sequel, for the purpose of theory, we will consider $\alpha_t = t^{-1}$, which corresponds to $\widehat{\mu}_t^{\mathsf{EMA}}$ being a flat average of the iterates $\theta_0, \ldots, \theta_t$, or, in continuous time:

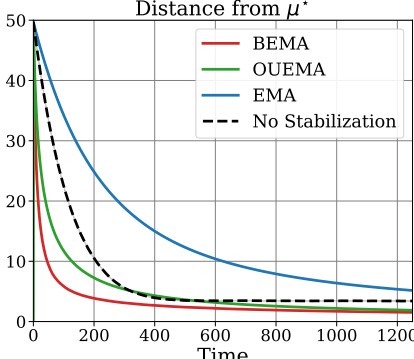

Distance from $\mu^\star$

Figure 2: Expected distance from the minimum $\mu^\star$ in stochastic quadratic optimization with $d = 20$ without stabilization, with EMA, BEMA, and OUEMA. EMA slows down the optimization process significantly, while BEMA and OUEMA converge significantly more quickly.

$$\widehat{\mu}_t^{\mathsf{EMA}} = \frac{1}{t} \int_0^t \theta_s \, ds. \tag{5}$$

A key advantage of $\widehat{\mu}_T^{\mathsf{EMA}}$ is that it is memory-efficient to compute in a streaming fashion; indeed, in order to return $\widehat{\mu}_T^{\mathsf{EMA}}$, a practitioner need only keep $\widehat{\mu}_t^{\mathsf{EMA}}$ in memory at any given time, along with the current iterate $\theta_t$. Once again, the simplicity of the OU process allows us to provide tight bounds on the expected squared error of this estimate.

**Proposition 3.** *Let $\theta_t$ be the solution to (2) with $\Sigma = \sigma^2 \mathbf{I}$ and let $\widehat{\mu}^{\mathsf{EMA}}$ be the estimator given in (5). Then*

$$\mathbb{E}\left[\left\|\widehat{\mu}_T^{\mathsf{EMA}} - \mu^\star\right\|^2\right] \le \frac{\eta\sigma^2 \cdot \mathrm{Tr}\left(\mathbf{A}^{-2}\right)}{T} + \frac{\left\|\mathbf{A}^{-1}\right\|_{\mathrm{op}}^2 \|\mu^\star - \theta_0\|^2}{T^2}. \tag{6}$$

*Moreover, if $T \le \frac{c}{2\lambda_{\max}(\mathbf{A})}$ for some constant $0 < c < 1$ then it holds that $\mathbb{E}\left[\left\|\widehat{\mu}_T^{\mathsf{EMA}} - \mu^\star\right\|^2\right] \ge (1 - c)^2 \|\mu^\star - \theta_0\|^2$.*

Comparing the upper bound in Proposition 3 to Proposition 1 implies that $\widehat{\mu}^{\mathsf{EMA}}$ is *asymptotically* optimal as $T$ grows, but the lower bound demonstrates that the higher order bias term in (6) is problematic for small $T$ when $\theta_0$ and $\mu^\star$ are not close to each other. In the context of optimization, it is reasonable to assume that $\theta_0$ and $\mu^\star$ are far apart (otherwise minimal optimization would

---

**Algorithm 1:** Bias corrected Exponential Moving Average (BEMA)

---

**Input:** Trajectory $\{\theta_t | t \in [T]\}$, EMA power $\kappa$, bias power $\eta$, multiplier $\gamma$, lag $\rho$, burn in time $\tau$, frequency $\phi$.

**Set** $\widehat{\mu} \leftarrow \theta_0$ and $\widehat{\mu}^{\text{EMA}} \leftarrow \theta_0$.

**for** $t = 1, \ldots, T$ **do**

   **if** $t \leq \tau$ **then**

      ⌊ **Update** $\widehat{\mu} \leftarrow \theta_t$, $\theta_0 \leftarrow \theta_t$, and $\widehat{\mu}^{\text{EMA}} \leftarrow \theta_t$.          (%% No bias correction before $\tau$)

   **else if** $(t - \tau) \mod \phi \neq 0$ **then**

      ⌊ **Continue**          (%% Only update every $\phi$ steps)

   **else**

      **Define** $\alpha_t \leftarrow (\rho + \gamma t)^{-\eta}$ and $\beta_t \leftarrow (\rho + \gamma t)^{-\kappa}$     (%% Define weights for EMA and bias correction.)

      **Update** $\widehat{\mu}^{\text{EMA}} \leftarrow (1 - \beta_t) \cdot \widehat{\mu}^{\text{EMA}} + \beta_t \cdot \theta_t$          (%% Update EMA.)

      **Update** $\widehat{\mu} \leftarrow \alpha_t (\theta_t - \theta_0) + \widehat{\mu}^{\text{EMA}}$

**Return** $\widehat{\mu}$.

---

be required), suggesting that $\widehat{\mu}^{\text{EMA}}$ suffers from significant bias when $\lambda_{\min}(\mathbf{A})T \lesssim 1$. This bias manifests itself as *lag* in the optimization process, which explains why the EMA curve in Figure 2 decreases more slowly than the vanilla optimization without stabilization.

One approach to improving upon $\widehat{\mu}^{\text{EMA}}$ is to debias the trajectory $\theta_t$ in a *pointwise* manner, i.e., introducing a new trajectory $\bar{\theta}_t$ such that for each $t$, $\mathbb{E}_{\mu^\star}[\bar{\theta}_t] = \mu^\star$; then we could apply iterate averaging to the augmented trajectory $\bar{\theta}_t$ and hopefully obtain a better estimator. The following result details the benefit of this approach.[3]

**Theorem 1.** *Let* $(\theta_t)_{0 \leq t \leq T}$ *be a trajectory from* (11) *with* $\mathbf{\Sigma} = \sigma^2 \mathbf{I}$ *and for some* $\tau \in (0, T)$, *let*

$$\widehat{\mu}_T^{OUEMA} = \frac{1}{T - \tau} \int_\tau^T \bar{\theta}_t dt \quad with \quad \bar{\theta}_t = \left(\mathbf{I} - e^{-\mathbf{A}t}\right)^{-1} \left(\theta_t - e^{-\mathbf{A}T}\theta_0\right).$$

*Then it holds for all* $t$ *that* $\mathbb{E}_{\mu^\star}[\bar{\theta}_t] = \mathbb{E}_{\mu^\star}[\widehat{\mu}_T^{OUEMA}] = \mu^\star$ *and*

$$\mathbb{E}\left[\left\|\widehat{\mu}_T^{OUEMA} - \mu^\star\right\|^2\right] \leq \frac{\eta \sigma^2 \cdot \text{Tr}\left(\mathbf{A}^{-2}\right)}{T\left[\left(1 - e^{-\lambda_{\min}(\mathbf{A})\tau}\right)(1 - \tau/T)\right]^2}. \tag{7}$$

Like $\widehat{\mu}^{\text{EMA}}$, the estimator $\widehat{\mu}^{\text{OUEMA}}$ can be implemented in a streaming fashion, as $\bar{\theta}_t$ is easy to compute given $\theta_t$; indeed, a practitioner need only hold in memory $\theta_0$, $\bar{\theta}_t$, and $\widehat{\mu}_t^{\text{OUEMA}}$ at any given time. Furthermore, in the regime where $\lambda_{\max}(\mathbf{A})T \lesssim 1$, when $\|\mu^\star - \theta_0\|$ is large relative to the conditioning of $\mathbf{A}$, it holds that $\widehat{\mu}^{\text{OUEMA}}$ improves upon $\widehat{\mu}^{\text{EMA}}$; indeed, this can lead to accelerated optimization of OUEMA relative to EMA, as can be seen in Figure 2.

While acceleration for small $T$ is a key benefit of $\widehat{\mu}^{\text{OUEMA}}$, this estimator is not asymptotically optimal, as can be seen by comparing the upper bound in (7) to the lower bound in Proposition 1. Thus, we instead propose $\widehat{\mu}^{\text{MLE}}$, the *Maximum Likelihood Estimator* of $\mu^\star$ as a candidate stabilizing algorithm (the practical instantiation of which is our main proposed intervention, BEMA).

**Theorem 2.** *Let* $\theta_t$ *be the solution to* (2) *with* $\mathbf{\Sigma} = \sigma^2 \mathbf{I}$. *Then*

$$\widehat{\mu}_T^{MLE} = \frac{\mathbf{A}^{-1}}{T} (\theta_T - \theta_0) + \frac{1}{T} \int_0^T \theta_t \, dt \tag{8}$$

*is the maximum likelihood estimator of* $\mu^\star$ *given the trajectory* $(\theta_t)_{0 \leq t \leq T}$. *Furthermore, it holds that* $\widehat{\mu}_T^{MLE}$ *is unbiased and* $\mathbb{E}\left[\left\|\widehat{\mu}^{MLE} - \mu^\star\right\|^2\right] = \frac{\sigma^2 \eta \cdot \text{Tr}\left(\mathbf{A}^{-2}\right)}{T}$.

The proof of Theorem 2 is deferred to Appendix E and crucially relies on Girsanov's theorem to express the log-likelihood of the process $\theta_t$ as a function of $\mu^\star$. Note that like $\widehat{\mu}^{\text{OUEMA}}$, the new

---

[3]See Appendix E.4 for a remark on why we apply a flat average here as opposed to a weighted average.

estimator $\widehat{\mu}^{\mathsf{MLE}}$ can be computed in a streaming fashion, requiring keeping in memory only two copies of the parameter at a time in addition to $\theta_t$. Moreover, $\widehat{\mu}^{\mathsf{MLE}}$ is optimal even in finite time, as can be seen by comparing the first term of the above bound to Proposition 1; indeed, we can even show that $\widehat{\mu}^{\mathsf{MLE}} - \mu^\star$ is distributed precisely as a Gaussian with covariance $\eta\sigma^2/T \cdot \mathbf{A}^{-2}$ (Corollary 1). In the regime where $T \cdot \lambda_{\max}(\mathbf{A}) \lesssim 1$, we expect $\widehat{\mu}^{\mathsf{MLE}}$ to significantly improve upon $\widehat{\mu}^{\mathsf{EMA}}$ whenever $\mu^\star$ and $\theta_0$ are far apart.

While thus far we have assumed that $\mathbf{A}$ is known, this assumption is unlikely to hold in the context of LM training, as $\mathbf{A}$ corresponds to the Hessian of the loss function, which is expensive to compute. In Appendix E, we prove Theorem 5, which controls the extent to which performance degrades when plugging in an incorrect $\widetilde{\mathbf{A}}$ into (8) in the place of $\mathbf{A}$. With respect to asymptotic in $T$ performance, the choice of $\widetilde{\mathbf{A}}$ is irrelevant, as it does not affect the leading term in the error bound, and the higher order terms in the resulting bound suggest that as long $\widetilde{\mathbf{A}}$ is not too far from $\mathbf{A}$ and not too poorly conditioned, the resulting estimator will still perform well. In practice, we find that taking $\widetilde{\mathbf{A}} = \alpha\mathbf{I}$ for some $\alpha > 0$ works well.

To complement the above theory, we visualize trajectories of a base OU process, EMA, and BEMA (the practical instantiation of $\widehat{\mu}^{\mathsf{MLE}}$) in Figure 1(a) in two dimensions with $\mu^\star = 0$, where we see that the base process jumps around due to the relatively large value of $\eta$, while EMA converges slowly because $\theta_0$ is relatively far from $\mu^\star$; on the other hand, BEMA converges significantly more quickly than the other two, in line with theory. A more involved comparison can also be seen in Figure 2, where we compare the expected squared distance between different estimators and $\mu^\star$ in stochastic quadratic optimization with $d = 20$ and $\mathbf{A} = \mathbf{I}$. We see that for this setting, EMA significantly delays convergence, while OUEMA and BEMA converge significantly more quickly, with BEMA being the fastest, as predicted by the continuous time theory once again.

## 4    PRACTICAL CONSIDERATIONS: INTRODUCING BEMA

Above, we established rigorous theoretical guarantees for a number of stabilizers under the assumption that $\theta_t$ is evolving as if it were an OU process. While these results can provide practical insight (Gupta et al., 2018; Vyas et al., 2024; Cohen et al., 2021; Block et al., 2024), in reality it is obviously not the case that when finetuning LMs, the loss landscape is exactly quadratic. Thus, in this section we describe our recommended intervention, BEMA. **We emphasize that BEMA only requires a two line change to existing EMA implementations, making it an easy, drop-in replacement.**

We are interested in finetuning LMs for two reasons: first, the combination of the desire to take as many gradient steps as possible with limited high-quality data necessitate using small batch sizes, which require stabilization; second, recent empirical work (Malladi et al., 2023) has suggested that post-training LMs can be well-approximated by a kernel (i.e. linear) setting due to the feature learning ocurring during pre-training, which suggests that the strong modelling assumptions made in Section 3 may be more reasonable. Even so, we must make several modifications to the theoretical estimator proposed in (8); pseudocode for the practical implementation, which we call BEMA, is given in Algorithm 1 (differences from standard EMA are colored in red). A summary of default hyperparameters is given in Table 2 in Appendix C.

First, a key difference between the theory of Section 3 and practice is that we must apply (8) in discrete time. This manifests in two ways: (a) we need to choose a value for the time $T$; (b) we need to implement the average comprising the second term of the estimator. Both issues already arise in existing implementations of EMA for deep learning (Izmailov et al., 2018; Busbridge et al., 2023; Block et al., 2024) and so we draw on popular, pre-existing solutions. Thus, instead of taking a flat average, we run an EMA with a constant that decays polynomially in the number of steps, with exponent $\kappa \in (0, 1)$; note that $\kappa = 1$ leads to a flat average, but practitioners have found that $\kappa < 1$ is often superior (Karras et al., 2023; Lee et al., 2024; Li et al., 2024; Wang, 2024).

Second, in order to respect the nonconvexity of the loss landscape and in line with empirical best practice, we allow a burn-in time $\tau$ before applying stabilization (either EMA or BEMA), although we find that $\tau = 0$ works best in practice when finetuning models. Third, while the iterate average can be quickly computed in a streaming fashion, the computational cost of updating BEMA at every step (which requires copying of model weights from one device to another in the likely event that CPU offloading is used to store $\theta_0$) can slow down training with respect to wall clock time. Thus

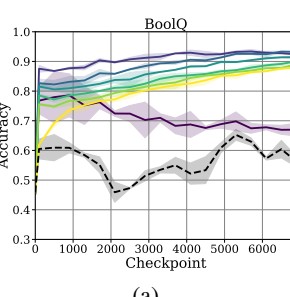 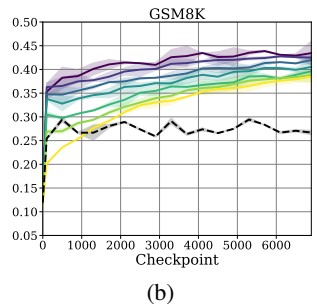 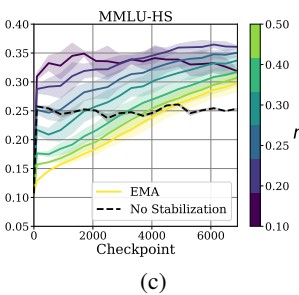

Figure 3: Effect of varying the $\eta$ hyperparameter in BEMA while finetuning Qwen2.5-1.5B on BoolQ (a), GSM8K (b), and MMLU-HS (c). In general, as $\eta$ decreases, the BEMA intervention over vanilla EMA gets stronger, leading to better performance. For too strong interventions ($\eta = 0.1$ in BoolQ), performance collapses, likely as a result of the failure of the quadratic approximation. In all cases, BEMA with $\eta = 0.2$ outperforms vanilla EMA.

we introduce a parameter $\phi$, governing the frequency with which we update our stabilizer; ideally $\phi$ is set so as to be large enough to provide computational savings but small enough to ensure local convexity of the loss landscape. We find that $\phi = 400$ works well in practice.

Finally, a practical consideration unique to BEMA is the choice of $\mathbf{A}$. While in theory it would be natural to treat this as a nuisance parameter to be estimated and plugged in,[4] naïvely doing this is infeasible: because $\theta_t \in \mathbb{R}^d$ for some large $d$, the $d^2$-dimensional $\mathbf{A}$ would likely not fit in memory. Practical adaptive and preconditioned optimizers have taken a variety of approaches to this problem, and integrating these into BEMA is an interesting direction for future work, but we find empirically that taking $\mathbf{A} = \alpha_t \cdot \mathbf{I}$ suffices to provide good performance in practice for some time-dependent scaling factor $\alpha_t$, akin to the $\beta_t$ used in EMA and discussed above. We set $\alpha_t = (\rho + \gamma t)^{-\eta}$, where $\rho$ is a lag term, $\gamma$ is a multiplier, and $\eta$ determines the rate of decay: smaller $\eta$ leads to a stronger intervention of our algorithm relative to EMA.[5] Combining the above yields BEMA.

# 5 FINETUNING LANGUAGE MODELS WITH BEMA

## 5.1 EMPIRICAL SETUP

We focus on *post-training*, where small batch sizes necessitate stabilization. We finetune on the Tulu-3-SFT dataset (Lambert et al., 2024), one of the largest and highest quality opensource finetuning datasets for LMs. Unless otherwise specified, all results are on the *pre-trained* Qwen2.5-1.5B model (Team, 2024), a 1.5B parameter model known for its strong performance on a number of benchmarks. We also consider the pretrained Gemma3-1B and Llama3.2-1B models in Appendix D. All training hyperparameters are summarized in Table 1 in Appendix C; we reran each run twice with different seeds to ensure our results are robust to the stochasticity in training. All of our training runs were done on a single 80 Gb NVIDIA A100 GPU, while our evaluations were conducted on 40 Gb NVIDIA A100 GPUs. Further details and experiments are deferred to Appendices C and D.

**Evaluation.** We consider 5 metrics in order to demonstrate the broad efficacy of BEMA. First, we consider train and test loss, which is the cross-entropy of the model on the current train batch and a fixed set of 200 held-out sequences respectively. Second, we consider three benchmarks for language generation and reasoning. For each prompt in each task, we generate 50 responses with temperature 1 and compare the model's responses to the groundtruth answer in order to estimate the average accuracy of the model on that prompt, then average across prompts to get the final score for each task. We consider the following tasks: (i) BoolQ. We randomly select 64 fixed prompts from

---

[4]Normally one would use an orthogonalized approach to such plug-in estimators (Chernozhukov et al., 2018), but in our setting $\widehat{\mu}^{\mathsf{MLE}}$ is already orthogonalized so we could simply plug in an estimate.

[5]In the case that $\eta = \infty$, we recover the standard EMA stabilizer.

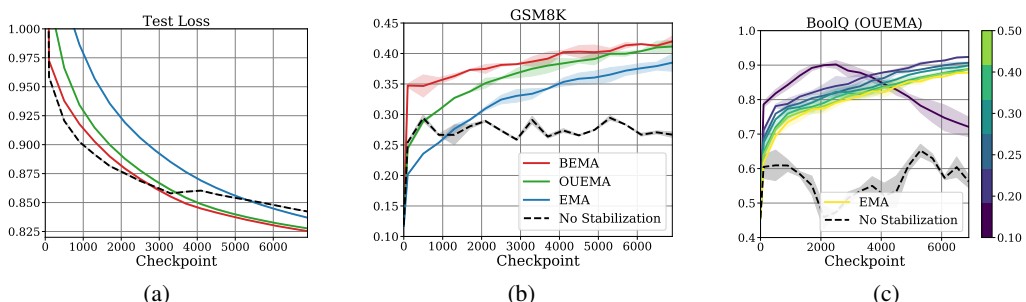

(a)                          (b)                          (c)

Figure 4: Performance of OUEMA as compared to BEMA on test loss **(a)**, GSM8K **(b)**, and demonstration of the effect of tuning the $\eta$ hyperparameter in OUEMA in BoolQ **(c)**. In all cases, BEMA outperforms OUEMA, which in turn outperforms vanilla EMA. As in the case of BEMA, making $\eta$ smaller (leading to a stronger intervention) in general leads to improved performance until the intervention becomes too strong and performance collapses.

the language understanding dataset BoolQ (Clark et al., 2019), which is a component of the standard SuperGLUE benchmark (Wang et al., 2019); (ii) GSM8K. We randomly select 128 fixed prompts from the test split of the standard mathematical reasoning dataset GSM8K (Cobbe et al., 2021); (iii) MMLU-HS. In order to satisfy constraints on computation, we evaluate on a strict subset of the MMLU dataset (Hendrycks et al., 2021). To ensure a fair and diverse selection of topics for which it is reasonable to expect good performance even for the relatively small models we consider, we select all of the topics labelled 'high school', leading to 14 topics detailed in Appendix C. For each topic, we randomly select 64 prompts for which to compute average model accuracy, then average across topics to get the final MMLU-HS score. We chose MMLU-HS and GSM8K due to the fact that they are both (subsets of) standard benchmarks for which improvement after finetuning on Tulu-3-SFT was demonstrated in Lambert et al. (2024). We chose BoolQ because it is a standard language understanding benchmark that is cheap to evaluate.

We emphasize that on all three tasks, by checking whether a model's generated text producess a (mostly) correctly formatted answer that matches the groundtruth correct answer, **we evaluate on generations, unlike many of the benchmark scores reported in the literature (Kamath et al., 2025; Grattafiori et al., 2024; Lambert et al., 2024) that turn models into classifiers** on multiple choice questions by selecting the answer with the highest probability; this difference explains why we report significantly lower scores for the pre-trained model as compared to the relevant technical reports. We discuss additional facets of our evaluation choices in Appendix C.

## 5.2 MAIN RESULTS

We now describe the main results of our empirical investigation. **Overall, we find that BEMA substantially improves upon both EMA and vanilla training on a diverse collection to tasks and models.** In Figure 1(b) we plot the training and test cross-entropy as a function of the number of gradient steps taken with $\kappa = 0.2$ (EMA power) and $\eta = 0.2$ (strength of BEMA correction) for vanilla training, EMA, and BEMA on Qwen2.5-1.5B; in both cases, we see that EMA leads to an initially slower convergence, although does ultimately outperform vanilla training in test loss, possibly due to the regularizing effects of early stopping (Prechelt, 2002; Yao et al., 2007) combined with the fact that EMA lag is functionally slowing down training. In both train and test losses, however, we see a considerable benefit of BEMA over EMA. A similar picture emerges in Figure 1(c), where we plot the performance with $\kappa = 0.5$ for BoolQ and GSM8K; in both cases, BEMA significantly outperforms EMA and vanilla training, both in peak performance and number of gradient steps required to achieve a given level of accuracy.

We provide further evidence for the empirical benefit of BEMA in Figure 3, where we examine the effect of varying $\eta$, with smaller $\eta$ leading to a stronger intervention of the bias correction term in BEMA. In each of the generations evaluation tasks we consider (BoolQ, GSM8K, and MMLU-HS), we see considerable gains of BEMA over mere EMA with $\eta = 0.2$ and this parameter appears fairly robust to the choice of task and other hyperparameters. While these results are for particular choices

of $\kappa$ and $\eta$, we explore plot corresponding training curves in Figure 5 for different values, as well as the optimal performance throughout training for several choices of $\kappa$ and $\eta$ in Figures 7 and 8 (cf. Appendix D). We observe that $\kappa = 0.5$ consistently leads to the best performance on downstream tasks, albeit with a significant increase in crossentropy that is somewhat mitigated by the bias correction term in BEMA. In Figure 4, we compare BEMA to OUEMA, a practical version of the estimator introduced in Theorem 1. We see that while OUEMA provides a significant improvement over EMA, it is consistently outperformed by BEMA, consistent with our theory.

We defer many ablations to Appendices A and D, wherein we demonstrate that BEMA continues to provide benefit across a wide range of optimizer hyperparameters, BEMA's hyperparameters, and models. We also compare BEMA to other stabilizers, including DEMA (Mulloy, 1994; Chen et al., 2023) (cf. Appendix C) and extensive tuning of OUEMA, finding that BEMA consistently outperforms both. We summarize these these further experiments in Appendix A.

## 6    DISCUSSION

In this work, we conceptualized the problem of designing *stabilizers* for stochastic optimization statistical estimation and proposed a new algorithm, BEMA, that is able to achieve the variance reduction advantages of EMA while at the same time providing accelerated convergence. Through theoretical analysis and extensive empirical evaluation, we demonstrated that **BEMA can provide substantial gains in finetuning language models, especially on their downstream performance, while being memory efficient and computationally inexpensive**.

**Related Work.**    A more complete survey of the literature can be found in Appendix B. Stochastic optimization has a long history, from the introduction of SGD (Robbins & Monro, 1951) through classical analyses of convex and quadratic settings, momentum, and iterate averaging (Nesterov, 2013; Ruppert, 1988; Polyak & Juditsky, 1992), with later work extending to nonconvex problems and refined variance bounds (Johnson & Zhang, 2013; Defazio et al., 2014; Schmidt et al., 2017; Défossez & Bach, 2015; Dieuleveut et al., 2017). Practical deep learning has layered on preconditioning (Duchi et al., 2011; Kingma & Ba, 2014; Gupta et al., 2018; Vyas et al., 2024), AdamW (Loshchilov & Hutter, 2017), and stabilizing interventions such as learning-rate decay, batch-size scaling, and iterate averaging (Izmailov et al., 2018; Sandler et al., 2023; Kaddour et al., 2023; Busbridge et al., 2023; Block et al., 2024). Of particular note are Défossez & Bach (2015); Dieuleveut et al. (2017), which analyze iterate-averaged SGD in the quadratic setting; the first focuses on asymptotic bounds and does not focus on the vanishing impact of the initial iterate (which is critical in the present work) while the second attains optimal bias (in discrete time, for high-dimensional problems, and up to constants). Neither considers the *stabilization* problem we introduce here. Most relevant to our work is Block et al. (2024), which introduced the notion of Gradient Variance Amplification (GVA) in the context of closed-loop evaluation of language models and observed the benefits of EMA in mitigating this phenomenon; our work builds on theirs by providing a theoretical framework for stabilizer design and proposing a new algorithm that improves upon EMA.

**Future Directions.**    In addition to further investigation as to the optimal choice of $\theta_0$ in BEMA, several immediate questions arise. First, we empirically instantiated BEMA with an isotropic prior, i.e., $\mathbf{A} = \mathbf{I}$, but it is natural to ask if some more adaptive choice could be made, e.g. using the existing second order information that AdamW and its variants provide. Second, our analysis arose out of the OU process, which is the scaling limit of SGD in quadratic optimization, but current LM training (including all of our experiments) uses adaptive methods; can we design stabilizers that are more directly linked to the scaling limit of these more sophisticated optimization algorithms? Third, we focused on the stabilizing process itself, treating the optimization trajectory as fixed; in practice, it may be even more effective to consider the question of *optimizer-stabilizer co-design*, wherein stabilization is considered to be a stochastic control problem as opposed to merely statistical estimation. In this case, we may find memory-efficient controllers that can adaptively accelerate convergence to the minimum while at the same time stabilizing the optimization trajectory, potentially leading to significant performance gains. Finally, our empirical focus was on SFT and did not consider alternative paradigms, such as Reinforcement Learning from Human Feedback (RLHF) (Christiano et al., 2017) or RL with Verifiable Rewards algorithms such as GRPO (Shao et al., 2024); exploring the effects of BEMA in these settings is an interesting direction for future work.

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

## REPRODUCIBILITY STATEMENT

We will make all code public to allow for reproducibility of our experiments upon publication. For now, note that the code involved in BEMA is a simple, two-line change to existing EMA implementations, of which there are many.[6] We provide full details of our experimental setup in Section 5 and Appendix C, including listing all training and stabilization hyperparameters in Tables 1 and 2. The proofs of all theoretical claims are provided in Appendix E.

## LARGE LANGUAGE MODEL USAGE

We used LLMs to help accelerate the production of code for our experiments, especially with respect to boilerplate training code. In addition, we used LLMs to help proofread and edit the writing in this paper. All code and writing suggestions were carefully reviewed and edited by the authors before inclusion in the final draft.

---

[6]While we used the popular HuggingFace library, a simple example of an open-source PyTorch implementation of EMA can be found in the github repository at https://github.com/lucidrains/ema-pytorch.

CONTENTS

## A  ADDITIONAL ABLATIONS

We now proceed to briefly describe the results of a number of ablations and further experiments that we performed, with detailed descriptions and results deferred to Appendix D.

**Does BEMA work with different optimizer hyperparameters?**    The results discussed so far are concordant with the theory presented in Section 3 insofar as training is conducted with a fixed learning rate. However, practitioners often observe a benefit of decaying the learning rate due to the improved stability that comes of decreasing the noise floor of stochastic optimization. Thus, we investigate in Figures 6 to 8 the effect of decaying the learning rate to zero and to $0.3$ times the peak learning rate on EMA and BEMA. Even with learning rate decay, both EMA and BEMA continue to exhibit improvement across the board and BEMA continues to outperform EMA; moreover, **we observe that training with a fixed learning rate and then applying BEMA leads to the best performance throughout, providing preliminary evidence that applying post-hoc stabilization can obviate the need for learning rate decay in post-training**. We also demonstrate that BEMA's performance is robust to the choice of batch size in Figure 13.

**What are the effects of changing BEMA hyperparameters?**    In Figure 9, we investigate the effect of changing $\tau$ (burn-in time) so that $\theta_0$ is the model after 500 or 1K gradient steps and stabilizing begins thereafter. In our experiments, setting $\tau = 0$ is significantly superior and leads to by

far the best performance. It is natural to ask if a more sophisticated scheme for setting $\theta_0$, such as a very slow-moving EMA as is done in, e.g. Grill et al. (2020); Pagliardini et al. (2024) would lead to improvement, and we leave this interesting question to future work. In addition to examining the effect of changing $\tau$, we also investigate the effect that the lag parameter $\rho$ has on the performance of BEMA in Figure 10, finding minimal effect across two orders of magnitude.

Of the three parameters, $\phi$ (the frequency with which we update) is by far the most important and we investigate its effect in Figures 11 and 12. Increasing $\phi$ leads to a decrease in the time of evaluation, as we do not need to perform the BEMA update as frequently, and the referenced figures demonstrate it has a significant regularizing effect as well. Indeed, we see that increasing the frequency (making $\phi$ smaller) leads to significant acceleration in convergence of training loss, but sometimes at the cost of overfitting.

**Is BEMA competitive?**   In Figure 4 we compare BEMA to OUEMA, EMA, and vanilla training without stabilization on test loss, GSM8K, and BoolQ. In all cases, we see that OUEMA significantly improves over EMA, as well as vanilla training, but is worse than BEMA across the board. This observation is further validated in Figure 15. In Figure 14, we compare BEMA to the so-called Double EMA (DEMA) (Mulloy, 1994; Chen et al., 2023), detailed in Appendix D. While we find that DEMA considerably improves over EMA, it holds that BEMA arrives at better performance substantially more quickly on all generation tasks we consider.

**Does BEMA work on other models?**   In Figures 16 and 17, we apply the identical procedure to Gemma3-1B (Kamath et al., 2025) and Llama3.2-1B (Grattafiori et al., 2024). We continue to see improvement of BEMA over EMA and vanilla training in crossentropy in all cases. In the case of Gemma3-1B, we see substantial gains in BoolQ, especially when $\kappa = 0.0$ (no EMA), while in Llama3.2-1B we see more modest gains in downstream performance. In both cases, especially the latter, the performance of the models is significantly lower than that of Qwen2.5-1.5B, at least partly due to the fact that neither model is as good at following instructions and thus both have trouble returning correctly formatted answers.

## B   RELATED WORK

We now summarize some relevant related work in the areas of stochastic optimization, stochastic differential equations, and the statistical estimation thereof.

**Stochastic Optimization.**   The study of stochastic optimization is classical, dating back to the introduction of SGD (Robbins & Monro, 1951). The theory in the convex setting is also classical (Nesterov, 2013), with momentum and especially iterate averaging (Ruppert, 1988; Polyak & Juditsky, 1992) emerging as powerful tools in variance reduction and acceleration. More recent works have investigated questions of nonasymptotic optimality in this setting both in the convex and nonconvex cases (Johnson & Zhang, 2013; Defazio et al., 2014; Schmidt et al., 2017). In the quadratic setting in particular, Défossez & Bach (2015) considered asymptotic bounds on the performance of itererate-averaged SGD with momentum while noting the asymptotically vanishing impact of the initial iterate on convergence; while this is tolerable their setting, the focus of the present paper is on reducing the impact thereof. Of particular relevance is Dieuleveut et al. (2017), which analyzes the performance of iterate-averaged SGD and attains the optimal bias (in discrete time, for high-dimensional problems, and up to constants) (Nesterov, 2013); note that our analysis does not violate this lower bound because (a) it occurs in continuous time and (b) the theoretically optimal estimator uses knowledge of the matrix $\mathbf{A}$, which is absent from the first-order stochastic optimization setting.

In addition to the well-developed classical theory, there has been a plethora of work spanning the past decade on incorporating ideas from stochastic convex optimization into practical deep learning algorithms, especially in the context of preconditioning (Duchi et al., 2011; Kingma & Ba, 2014; Gupta et al., 2018; Vyas et al., 2024). While our theory does not directly address the question of pre-conditioned optimizers (nor does it incorporate momentum), all of our empirical results use the standard AdamW optimizer (Loshchilov & Hutter, 2017), which incorporates both. Many of these popular training interventions already function in a stabilizing role, including increasing the batch size, shrinking the learning rate, and using a more aggressive learning rate decay schedule (Blum, 1954; Krizhevsky et al., 2012), but these approaches all come with their own drawbacks:

increasing the batch size is often infeasible due to data scarcity, while smaller learning rates and more aggressive decay schedules can significantly slow training (Smith & Topin, 2019; Loshchilov & Hutter, 2022). Thus, one of the most empirically successful approaches to stabilizing training is iterate averaging, which plays a crucial tool both in achieving optimal algorithms in the theory of stochastic optimization (Ruppert, 1988; Polyak & Juditsky, 1992; Defazio et al., 2014; Johnson & Zhang, 2013), and in empirical deep learning, beginning with Izmailov et al. (2018). Many recent works have specifically investigated the effects of EMA in this context (Sandler et al., 2023; Kaddour et al., 2023; Kaddour, 2022; Busbridge et al., 2023). Most relevant to the present work is Block et al. (2024), which focused on the effect EMA has on evaluating closed-loop rollouts and introduced the notion of Gradient Variance Amplification (GVA). While that work primarily observed the benefits of EMA, our work is devoted to understanding ways to improve stabilizer design, formalized as a statistical estimation problem.

**Stochastic Differential Equations: Estimation and Optimization.**    The study of stochastic Differential Equations (SDEs) is classical and has a rich theory built around it (Liptser & Shiryaev, 2013a;a; Le Gall, 2016). In the context of optimization in deep learning, SDEs have been considered as a useful tool for analyzing scaling limits of the discrete optimization trajectory (Li et al., 2017; Malladi et al., 2023; Busbridge et al., 2023), which in turn help understand how to tune hyperparameters such as the learning rate and momentum when other training settings are changed. In addition to analyzing scaling limits, many works have analyzed generalization properties of stochastic optimization as sampling from an SDE (Raginsky et al., 2017; Ben Arous et al., 2022), with Mandt et al. (2015) in particular analyzing the OU process. While we make use of the continuous time limit of SGD for the sake of analysis, our focus is more on designing new stabilizers than on the precise scaling limits thereof.

A more classical question in SDEs than that of deep learning scaling limits is the problem of parameter estimation, where a learner observes a single trajectory from the solution to an unknown SDE and seeks to estimate the parameters thereof (Liptser & Shiryaev, 2013b; Kutoyants, 2013); typically, authors have focused on the infinite time limits of such questions, examining the asymptotic properties of a variety of estimators. Of particular relevance to the present work is the application of Maximum Likelihood Estimation (MLE) (Wilks, 1938) to this problem, with many authors considering the asymptotic performance and optimality of the MLE in a variety of settings (Kutoyants, 2013), including in the OU process. In addition, parameter estimation of stochastic processes (especially the OU process) has been extensively studied in the context of mathematical finance (cf. e.g. Zhang et al. (2014) and the references therein), although the focus of those works tend to involve estimation from discrete time observations rather than the full continuous time trajectory. In contradistinction to those works, our primary interest is in memory- and computationally-efficient estimators that can be practically deployed at the scale of modern LMs. While the form of the MLE has certainly been worked out in the literature, to the best of our knowledge, its application to stochastic optimization is novel to the present work.

## C    FURTHER DETAILS ON EMPIRICAL SETUP

In this section, we provide further details on our empirical setup, including the base training hyperparameters, precise statistics of the dataset we use, and our evaluation setup.

**Training Data.**    We use the Tulu-3-SFT dataset (Lambert et al., 2024), which ordinarily consists of about 1M sequences as our training data. We randomly split the data, keeping 99% for training and 1% for validation. We then filter our training set to ensure that each sequence has at most 4096 tokens as per the Qwen2.5-1.5B tokenizer in order to prevent any memory issues. This results in 929,949 distinct training sequences, amounting to about 600M tokens. We then randomly select 200 sequences from the validation set for evaluation during training, which we keep fixed throughout.

**Training.**    The default hyperparameters we use for training are summarized in Table 1. While by default we do not use any learning rate decay, as we find that after stabilization this leads to the best performance, when we do experiment with learning rate decay, we use a linear schedule without warmup, decaying to some constant fraction of the peak learning rate. We train using the HuggingFace transformers and trl libraries (Wolf et al., 2019; von Werra et al., 2020), in particular

using the SFTTrainer class. All training was conducted on 80 Gb NVIDIA A100 GPUs. When training Gemma3-1B and Llama3.2-1B models, we use their tokenizers but always enforce the chat template used for Qwen2.5-1.5B in an effort to ensure consistency. We ran for the 2 epochs recommended by (Lambert et al., 2024).

**Stabilizers.** We consider four candidate stabilizers: EMA, BEMA, OUEMA, and DEMA. The implementation of BEMA is given in Algorithm 1, with $\gamma = 1.0$ and, by default, $\rho = 10$. We implement EMA as a special case of BEMA, but with $\eta = \infty$, which removes the bias correction term. To implement OUEMA, we compute

$$\bar{\theta}_t = \left(1 - \frac{1}{(1 + \gamma t)^\eta}\right)^{-1} \left(\theta_t - \frac{1}{(1 + \gamma t)^\eta}\theta_0\right)$$

and then apply the EMA update rule in Algorithm 1, but with $\theta_t$ replaced by $\bar{\theta}_t$. Finally, we discuss DEMA in Appendix D. In all cases, we set $\gamma = 1.0$ and, unless otherwise specified, we set $\rho = 10$. Finally, unless otherwise noted, we always assume the frequency $\phi = 400$ in order to reduce the computational cost of evaluation. Default hyperparameters for the stabilizers are summarized in Table 2.

**Evaluation.** We evaluate the candidate stabilizers (EMA, BEMA, OUEMA, and DEMA) on saved checkpoints so as to escape the need to retrain the model multiple times. By default, in an effort to reduce computation, we update the stabilizer every 400 gradient steps, although we investigate the effect of this choice in Appendix D. As described in Section 5, we consider train and test losses, consisting of cross-entropy on the current training batch and a fixed set of 200 held-out sequences, respectively. We also consider BoolQ, GSM8K, and MMLU-HS as benchmarks for language generation and reasoning. We use vLLM (Kwon et al., 2023) to generate responses in each case, and we once again emphasize that our evaluation is on generations, requiring the model to not just know the correct answer to a given question, but also to (at least loosely) follow the formatting instructions given in the prompt so that we can parse the final answer. In an effort to reduce the computational cost of evaluation and in recognition of the relatively small size of the models we consider, we restrict MMLU to MMLU-HS, the set of high school topics, which we take to be the 'easy' topics in that benchmark. We use all such topics in order to avoid cherrypicking by subject. The topics are as follows: high_school_biology, high_school_chemistry, high_school_computer_science, high_school_european_history, high_school_geography, high_school_government_and_politics, high_school_macroeconomics, high_school_mathematics, high_school_microeconomics, high_school_physics, high_school_psychology, high_school_statistics, high_school_us_history, and high_school_world_history. For the generation evaluations (BoolQ, GSM8K, and MMLU-HS), we randomly select prompts and, for each prompt, generate 50 responses with temperature 1, computing the average accuracy per prompt in order to reduce the variance of our estimates of model quality. For GSM8K and MMLU-HS, we use Chain of Thought prompting (Wei et al., 2022) and for BoolQ we use the common choice of 5-shot prompting (Kamath et al., 2025; Grattafiori et al., 2024).

**Remark 1.** We emphasize that our evaluation procedure is on tasks for which the model is not directly trained, in that the only training the model receives is on the Tulu-3-SFT dataset, which is a general-purpose dataset intended to improve question-answering, reasoning, and instruction following. As such, there is no *a priori* reason that performance on BoolQ, GSM8K, or MMLU-HS should necessarily improve after training, although we do observe that it does. Some reason for this improvement is that the model learns to better follow the formatting instructions in the prompts, allowing answers to be correctly parsed. Recent discussion of the effect that correct formatting has on benchmark performance has emphasized that a number of approaches that claim to enhance reasoning abilities in LMs actually do so by improving the model's ability to follow formatting instructions (Chandak et al., 2025) and as such it is critical in such evaluations to ensure that the correct benchmark is used. Note that this is not a problem in our work as we are not claiming that BEMA improves reasoning abilities, but rather that it improves optimization; because Tulu-3-SFT is designed to improve the model's ability to follow formatting instructions the evaluations we conduct indeed measure that which we claim. To reiterate, the goal of our evaluation suite is not to lift performance on the benchmarks qua benchmark performance, but rather to measure the model's quality

Table 1: Default Training Hyperparameters

| Hyperparameter | Value |
|---|---|
| Model | Qwen2.5-1.5B |
| Tokenizer | Qwen2.5-1.5B |
| Epochs | 2 |
| Peak Learning Rate | $3 \times 10^{-5}$ |
| Effective Batch Size | 256 |
| Learning Rate Decay | None |
| Warmup Steps | 0 |
| dtype | fp16 |
| Optimizer | Adam |
| Adam $\beta_1$ | 0.9 |
| Adam $\beta_2$ | 0.999 |
| Adam $\varepsilon$ | $10^{-8}$ |
| Gradient Clipping | 1.0 |

Table 2: Default BEMA Hyperparameters

| Hyperparameter | Value |
|---|---|
| EMA Power $\kappa$ | 0.5 |
| Bias Correction Power $\eta$ | 0.2 |
| Multiplier $\gamma$ | 1.0 |
| Lag $\rho$ | 10 |
| Burn-in $\tau$ | 0 |
| Frequency $\phi$ | 400 |

when evaluated closed-loop in the precise regime that Gradient Variance Amplification (GVA) is a problem (Block et al., 2024).

## D  FURTHER EMPIRICAL RESULTS

We now describe in detail the additional empirical results and ablations we conducted that were briefly alluded to, but not extensively discussed, in the main text. In particular, we conduct a thorough and exhaustive investigation of the sensitivity of BEMA to its hyperparameters as well as those of training. We then compare BEMA to alternative stabilizers, such as OUEMA and DEMA, and conclude by evaluating BEMA on Gemma3-1B and Llama3.2-1B in order to ensure that our results are not specific to Qwen2.5-1.5B.

**Changing $\eta$ and $\kappa$.** In Figure 5, we display the train and test crossentropy losses as well as BoolQ performance for different values of $\kappa$, which tunes the strength of the EMA intervention, and different values of $\eta$, which tunes the strength of the bias correction. In general, we find that as $\kappa$ is increased (leaading to more aggressive averaging), the optimization trajectory can handle lower values of $\eta$ (stronger intervention of bias correction), with the maximal $\kappa$ we tried reaching the highest performance on a number of tasks. A common pathology for small values of $\kappa$ is that the cross entropy losses improves significantly over both EMA and vanilla optimization, but the BoolQ performance suffers, likely due to the misalignment between Tulu-3-SFT and BoolQ resulting in a form of overfitting to the SFT task. Indeed, for the highest performance of BEMA on all generations tasks, which is achieved with $\kappa = 0.5$, we find that the training and test crossentropy losses are substantially larger than those achieved with smaller $\kappa$, again pointing to the misalignment between SFT task and generation. It is clear, however, that BEMA imparts considerable advantage in terms of acceleration relative to EMA and vanilla optimization.

**Changing the learning rate through learning rate decay.** In Figures 6 to 8, we investigate the effect of learning rate decay on BEMA and EMA. In the latter two figures, we plot the optimal

throughout training losses in crossentropy (for both train and test sets) as well as performance on the considered generations tasks. We see that the optimal performance accross the board occurs *without any learning rate decay but with stabilization via* BEMA. That said, the effect learning rate decay has on the optimization trajectories without stabilizing, with EMA, and with BEMA can all be observed in Figure 6 and appears to be present, but small.

**Effect of BEMA hyperparameters.** In addition to the $\kappa$ and $\eta$ hyperparameters of BEMA described above, three other choices can potentially affect the performance of BEMA : (1) the choice of the burn-in time $\tau$; (2) the choice fo the lag $\rho$; and (3) the choice of update frequency $\phi$. In Figure 9, we demonstrate that, at least in our setup, choosing $\tau = 0$ (no burn-in) leads to by far the best performance, with waiting 500 or 1000 steps leading to substantial degradation. We conjecture that this is a general phenomenon when starting with pre-trained models and aligns with earlier work suggesting that post-training approximately occurs in a convex setting (Malladi et al., 2023). Beyond $\tau$, we see in Figure 10 that the choice of lag $\rho$ has minimal effect on the stabilization, which is unsurprising considering that after sufficiently many steps, the lag does not meaningfully affect the update itself.

Of these three hyperparameters, the update frequency $\phi$ is by far the most signifcant. In Figures 11 and 12 we investigate the effect that updating significantly more frequently has on EMA and BEMA. We find that for small values of $\phi$ (very frequent updates), the convergence of BEMA is considerably accelerated, leading to much lower train and test losses. The phenomenon whereby the model overfits to the SFT task and performance (after sufficient training) declines on BoolQ is significantly magnified by this acceleration as well. In all cases, however, we continue to see significant acceleration benefits of BEMA relative to EMA and vanilla optimization.

**Effect of batch size.** In Figure 13, we demonstrate that BEMA continues to provide significant acceleration after the batch size is doubled. Indeed, one might expect EMA to provide less benefit in this case due to the reduction of stochasticity in the gradients, but the factor of 2 increase to an effective batch size of 512 does not appear to impact the performance overmuch and we continue to see gains from BEMA relative to EMA and vanilla optimization.

**Comparison to alternative stabilizers.** In Figure 15, we display plots analogous to those of Figure 5, but for OUEMA instead of BEMA. We vary $\kappa$ from removing all averaging up to $\kappa = 0.5$ and consider a number of values of $\eta$, comparing the performance of OUEMA to that of vanilla optimization, EMA, and BEMA with tuned $\eta$ value. We find that BEMA signifcantly outperforms OUEMA accross the board, and OUEMA tends to outperform EMA with respect to acceleration.

In addition to comparing BEMA to OUEMA, we also consider the Double Exponential Moving Average (DEMA), which updates according to the following rule:

$$\theta_t^{\mathsf{DEMA}} = 2 \cdot \theta_t^{\mathsf{EMA}} - \theta_t^{\mathsf{EMA,EMA}}$$
$$\theta_t^{\mathsf{EMA}} = (1 - \beta_t) \cdot \theta_{t-1}^{\mathsf{EMA}} + \beta_t \cdot \theta_t$$
$$\theta_t^{\mathsf{EMA,EMA}} = (1 - \beta_t) \cdot \theta_{t-1}^{\mathsf{EMA,EMA}} + \beta_t \cdot \theta_t^{\mathsf{EMA}},$$

i.e., $\mathsf{DEMA} = 2 \cdot \mathsf{EMA} - \mathsf{EMA}\,(\mathsf{EMA})$. This stabilizer comes out of the finance literature (Mulloy, 1994) and has recently been applied to training neural networks as an alternative to EMA (Chen et al., 2023). In Figure 14, we compare DEMA to BEMA and OUEMA on the quadratic optimization problem of Figure 2, the crossentropy losses, and the generations losses BoolQ, GSM8K, and MMLU-HS. In the quadratic case, we see that DEMA initially improves on EMA, but then eventually matches the performance thereof, and is uncompetitive relative to BEMA, as the theory predicts. In the case of crossentropy, DEMA improves on OUEMA and EMA, but has inferior training loss to BEMA, and less acceleration than the same in terms of test loss. Finally, BEMA continues to outperform DEMA on the generations tasks, leading to substantial acceleration and sometimes superior peak performance.

**Performance on Gemma3-1B and Llama3.2-1B.** Finally, in order to ensure that our results are not specific to Qwen2.5-1.5B, we also evaluate BEMA on Gemma3-1B and Llama3.2-1B in Figures 16 and 17. We find that BEMA continues to provide significant acceleration relative to EMA on train and test crossentropy losses in both models, as well as in the generations tasks in

the default setup with $\kappa = 0.5$. On the other hand, in several of these examples, we find that vanilla optimization without stabilization actually outperforms both EMA and BEMA, especially with Llama3.2-1B; in Gemma3-1B with BoolQ, however, we continue to see gains. Further investigation revealed that Gemma3-1B and especially Llama3.2-1B continue to have problems following instructions, leading to wrong answers by default, even after finetuning on Tulu-3-SFT; thus while these results are in general encouraging for BEMA, a more complete evaluation on these models with a different suite of tasks that is more commensurate to their capabilities is necessary to firm up these conclusions, which we leave for future work. We conclude by noting that even without averaging, i.e., setting $\kappa = 0$, leads to significant improvements in BoolQ performance when using BEMA over vanilla training without stabilization, especially for Gemma3-1B.

# E ADDITIONAL THEORETICAL RESULTS AND PROOFS

In this appendix, we provide formal proofs of the results in the main text. We begin by proving several elementary facts about the Ornstein-Uhlenbeck process and general diffusions, as well as the lower bound for well-behaved estimators. We then prove several results about $\widehat{\mu}^{\mathsf{MLE}}$ as consequences of a general theorem and then conclude by proving upper bounds on the performance of $\widehat{\mu}^{\mathsf{OUEMA}}$.

**Notation.** We will use $\|\cdot\|$ to denote the Euclidean norm in $\mathbb{R}^d$ and $\|\cdot\|_{\mathrm{op}}$ and $\|\cdot\|_{\mathrm{F}}$ to denote the operator and Frobenius norms of matrices, respectively. We will let $\mathbf{I}$ be the identity matrix and reserve bold capital letters for matrices; the trace of a matrix is denoted by $\mathrm{Tr}(\cdot)$. Given random vectors $a, b$, we denote $\mathrm{Cov}(a, b) = \mathbb{E}\left[ab^\top\right] - \mathbb{E}[a]\mathbb{E}\left[b^\top\right]$ the covariance matrix, and abbreviate $\mathrm{Cov}(a) = \mathrm{Cov}(a, a)$; furthermore, we let $\mathrm{Var}(a) = \mathrm{Tr}(\mathrm{Cov}(a)) = \mathbb{E}\left[\|a - \mathbb{E}[a]\|^2\right]$.

## E.1 TECHNICAL PRELIMINARIES

We begin by recalling a version of the classic Girsanov theorem, which is indispensible for our analysis of the Maximum Likelihood Estimator. For more details on generalizations and applications of Girsanov's theorem, we refer the reader to Le Gall (2016); Liptser & Shiryaev (2013a). The form of this result we use is as follows.

**Theorem 3** (Girsanov's Theorem). *Let $f : \mathbb{R}^d \to \mathbb{R}$ be a differentiable function and suppose that*

$$\mathbb{E}\left[\sqrt{\int_0^T \left\|\mathbf{\Sigma}^{-1}\nabla f(W_t - \mu)\right\|^2 dt}\right] < \infty, \tag{9}$$

$$\mathbb{E}\left[\exp\left(\frac{1}{2}\int_0^T \mathbf{\Sigma}^{-1}\nabla f(W_t - \mu)dW_t\right)\right] < \infty. \tag{10}$$

*Then $\theta^\mu$, the solution to*

$$d\theta_t^\mu = -\nabla f(\theta_t^\mu - \mu)dt + \mathbf{\Sigma}dW_t, \quad \theta_0^\mu = \theta_0,$$

*exists and*

$$\frac{d\mathbb{P}^\mu}{d\mathbb{P}^W}((\theta_t)_{0 \le t \le T}) = \exp\left(-\int_0^t \left\langle\mathbf{\Sigma}^{-2}\nabla f(\theta_s^\mu - \mu), d\theta_s^\mu\right\rangle - \frac{1}{2}\int_0^t \left\|\mathbf{\Sigma}^{-1}\nabla f(\theta_s^\mu - \mu)\right\|^2 ds\right),$$

*where $\mathbb{P}^W$ is the Wiener measure.*

*Proof.* By Le Gall (2016, Corollary 5.17), (9) implies that the process $L_t = \int_0^t \nabla f(W_s - \mu)dW_s$ is a uniformly integrable martingale for $t \in [0, T]$. As (10) is precisely Kazamaki's condition (cf. Le Gall (2016, Theorem 5.23)), Girsanov's theorem (Le Gall (2016, Theorem 5.22)) implies the result. A one-dimensional version of this result is also given, e.g., in Kutoyants (2013, Theorem 1.12). $\qquad\square$

**Remark 2.** Recall that Novikov's condition (cf. Le Gall (2016); Liptser & Shiryaev (2013a)) is sufficient to ensure that the conclusion of Theorem 3 holds and is often easier to verify than (9) and (10). Unfortunately, for our main application below, that of an OU process, for Novikov's condition

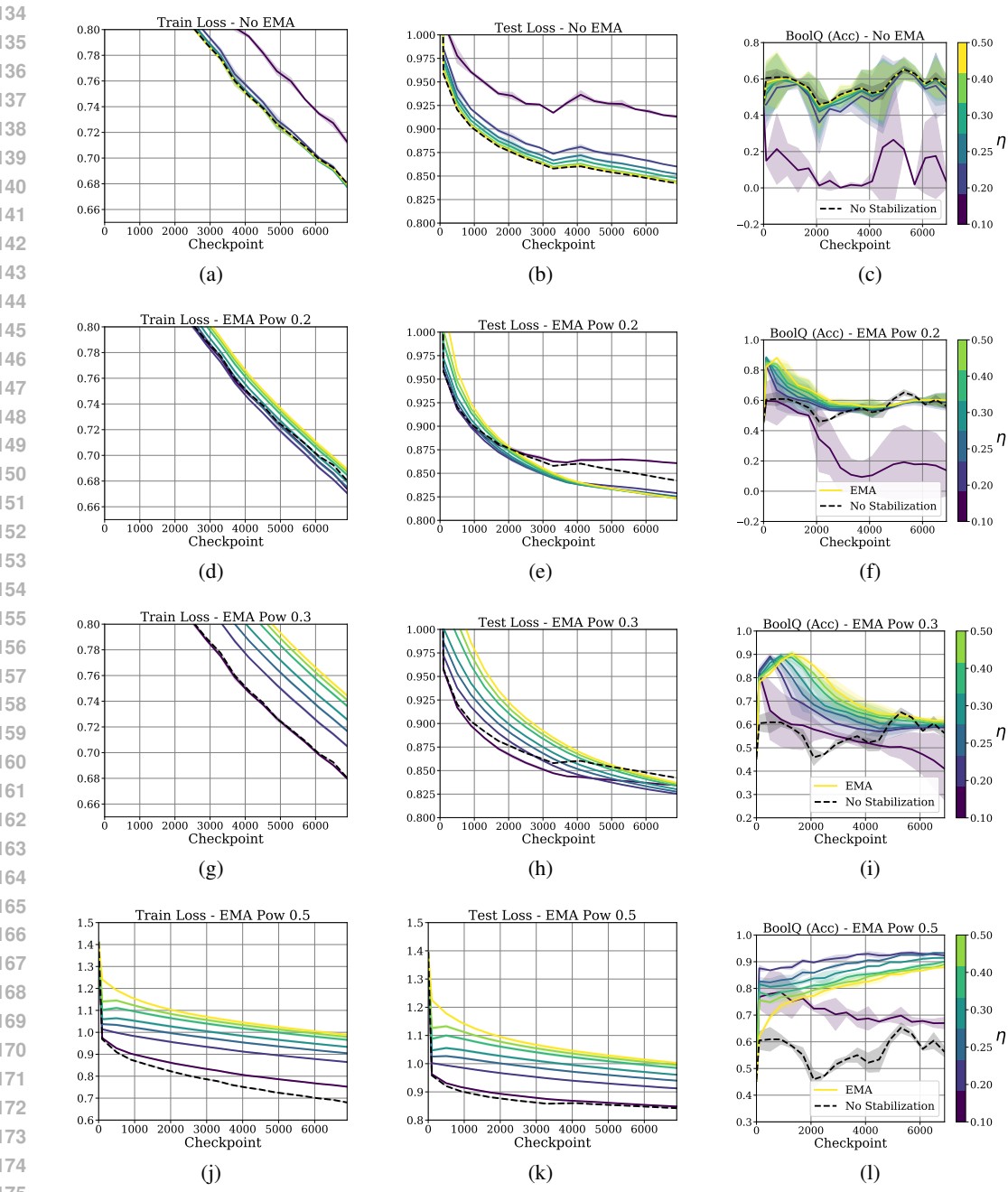

Figure 5: Performance of BEMA with $\kappa = 0.0$ (no EMA) and other values $\kappa$ with respect to **(first column)** train loss, **(second column)** test loss, and **(third column)** BoolQ accuracy. BEMA performance generally increases with $\kappa$ as training allows for lower values of $\eta$, leading to a stronger intervention of BEMA over EMA.

to hold we would require $\Sigma^{-1}A \prec 2\sqrt{\eta} \cdot I$, with $\Sigma$ and $A$ as in (2). As this is unnecessarily restrictive, we instead apply *Kazamaki's Criterion* (cf. Le Gall (2016, Theorem 5.23)), which is a more general condition that is satisfied by the OU process and thus allows us to apply Girsanov's theorem in this case.

We now apply this result to the OU process explicitly.

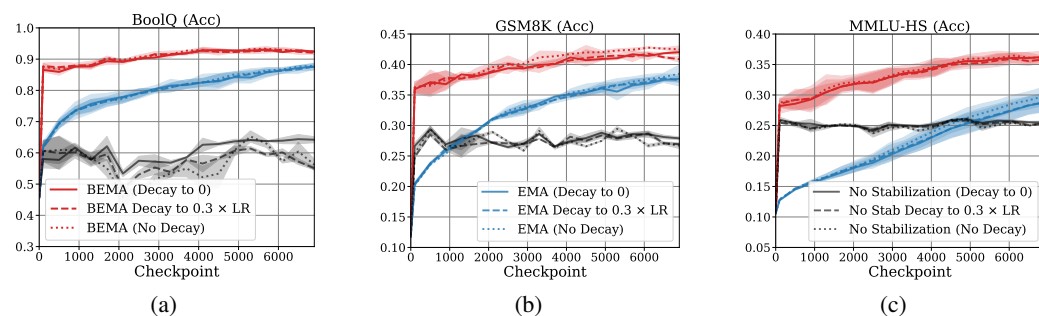

Figure 6: Demonstration of the effect of learning rate decay on training with stabilization, both with EMA and BEMA. Evaluations on **(a)** BoolQ, **(b)** GSM8K, and **(c)** MMLU-HS suggest that BEMA robustly improves on EMA performance for a range of learning rates.

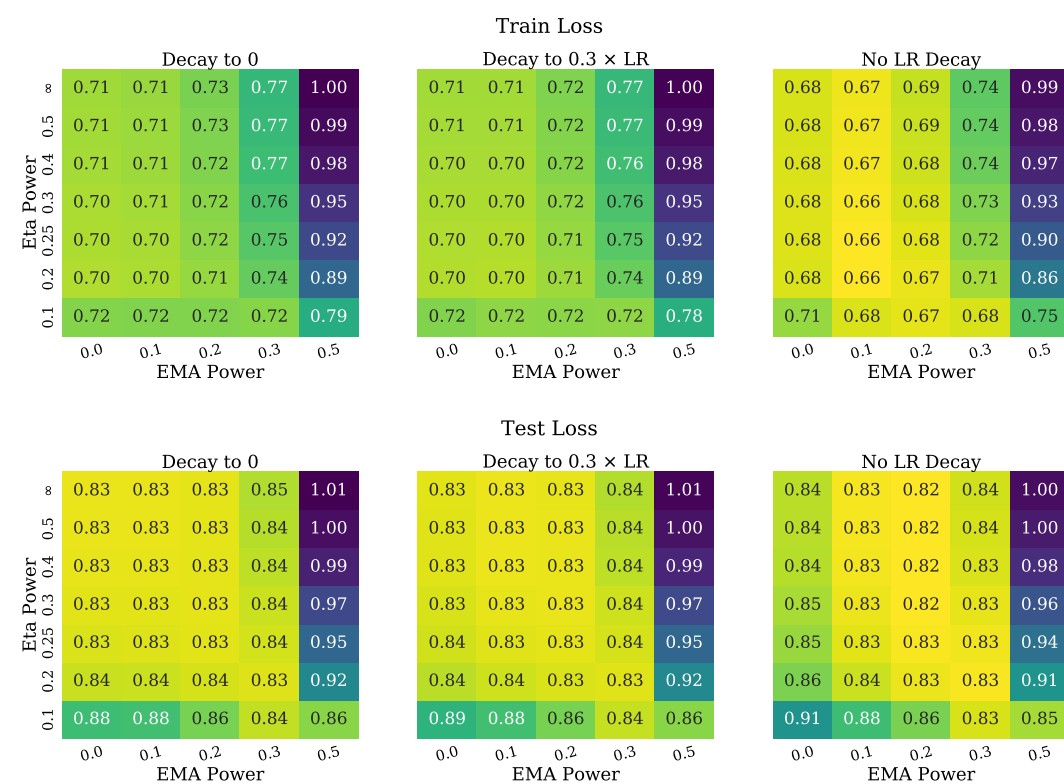

Figure 7: Effect of $\kappa$ and $\eta$ on best over training trajectory train **(top)** and test **(bottom)** loss for BEMA for decay to 0 **(left)**, decay to 0.3 times peak learning rate **(middle)**, and no decay **(right)**.

**Proposition 4.** *Let $(\theta_t)_{0 \le t \le T}$ be the solution to the OU process* (2) *with $\mathbf{A}, \boldsymbol{\Sigma} \in \mathbb{R}^{d \times d}$ symmetric positive definite and let $\mathbb{P}^{\mu^\star}$ denote the measure of paths under this law. If $\mathbb{P}^W$ is the Wiener measure, then it holds that*

$$\log \frac{d\mathbb{P}^{\mu^\star}}{d\mathbb{P}^W}(\theta_t) = -\frac{1}{\eta} \int_0^T \left\langle \boldsymbol{\Sigma}^{-2}\mathbf{A}(\mu^\star - \theta_t), d\theta_t \right\rangle - \frac{1}{2\eta} \int_0^T \left\| \boldsymbol{\Sigma}^{-1}\mathbf{A}(\mu^\star - \theta_t) \right\|^2 \, dt.$$

*Proof.* We apply Theorem 3 with $f(\theta) = \frac{1}{2}\theta^\top \mathbf{A}\theta$. Note that

$$\nabla f(\theta - \mu^\star) = \mathbf{A}(\mu^\star - \theta) \quad \text{and} \quad \nabla^2 f(\theta - \mu^\star) = \mathbf{A}.$$

Replacing $\boldsymbol{\Sigma}$ by $\sqrt{\eta} \cdot \boldsymbol{\Sigma}$ in Girsanov's theorem above yields the result, given that (9) and (10) hold. Thus it remains to establish these inequalities. The first inequality holds by Holder, the linearity of

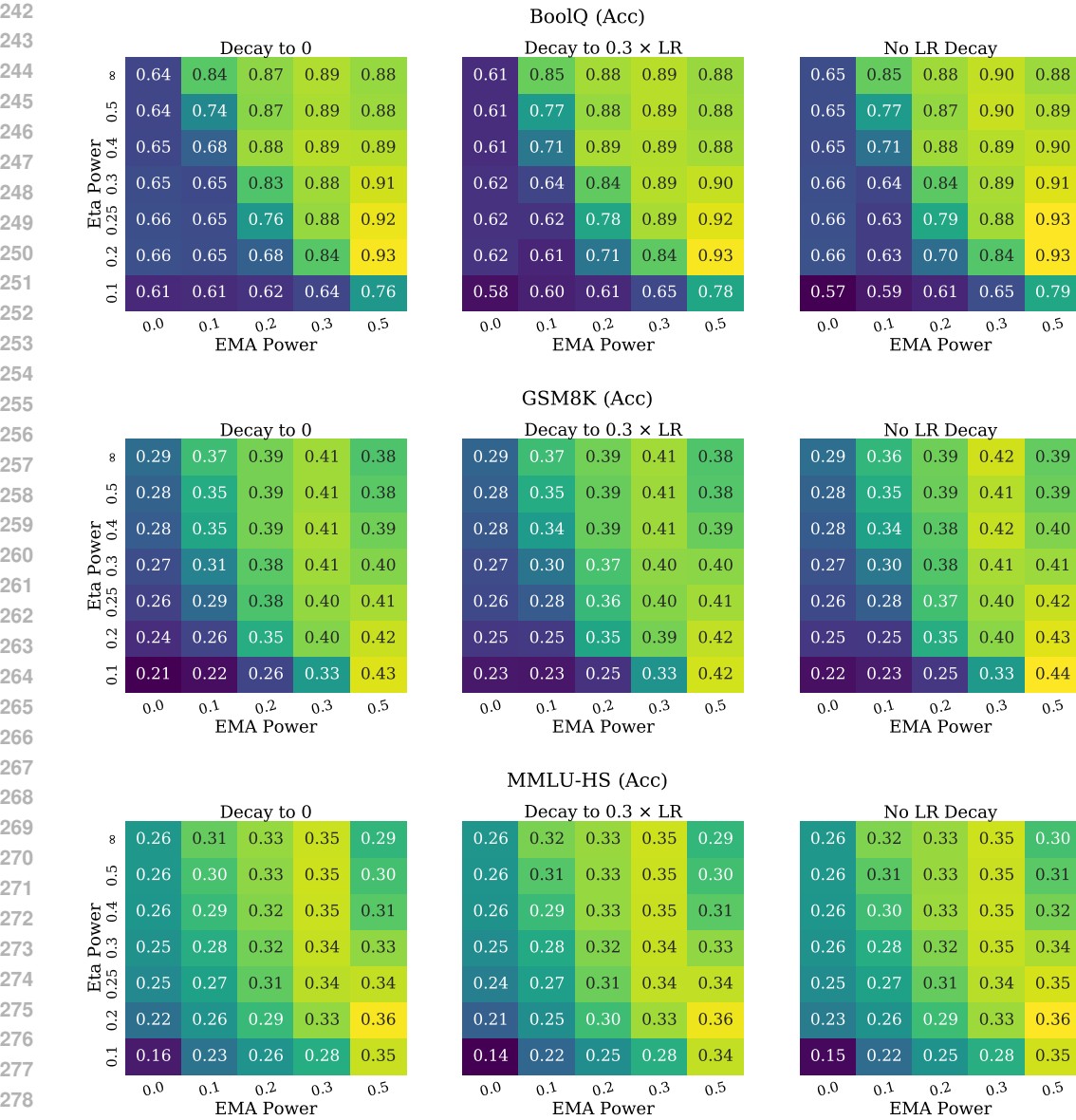

Figure 8: Effect of $\kappa$ and $\eta$ on optimal throughout training performance on BoolQ **(top)**, GSM8K **(middle)**, and MMLU-HS **(bottom)** for BEMA for decay to 0 **(left)**, decay to 0.3 times peak learning rate **(middle)**, and no decay **(right)**. Compared to pure EMA ($\eta = \infty$), BEMA not only accelerates convergence, but can lead to better performance in the long run.

expectation, and the fact that Gaussians have finite second moments:

$$\mathbb{E}\left[\sqrt{\int_0^T \|\mathbf{\Sigma}^{-1}\nabla f(W_t - \mu)\|^2\, dt}\right] \leq \sqrt{\mathbb{E}\left[\int_0^T \|\mathbf{\Sigma}^{-1}\nabla f(W_t - \mu)\|^2\, dt\right]}$$

$$= \sqrt{\int_0^T \mathbb{E}\left[\|-\mathbf{\Sigma}^{-1}\mathbf{A}W_t\|^2\right] dt}$$

$$= \sqrt{T \cdot \mathrm{Tr}\left(\mathbf{\Sigma}^{-2}\mathbf{A}^2\right)} < \infty.$$

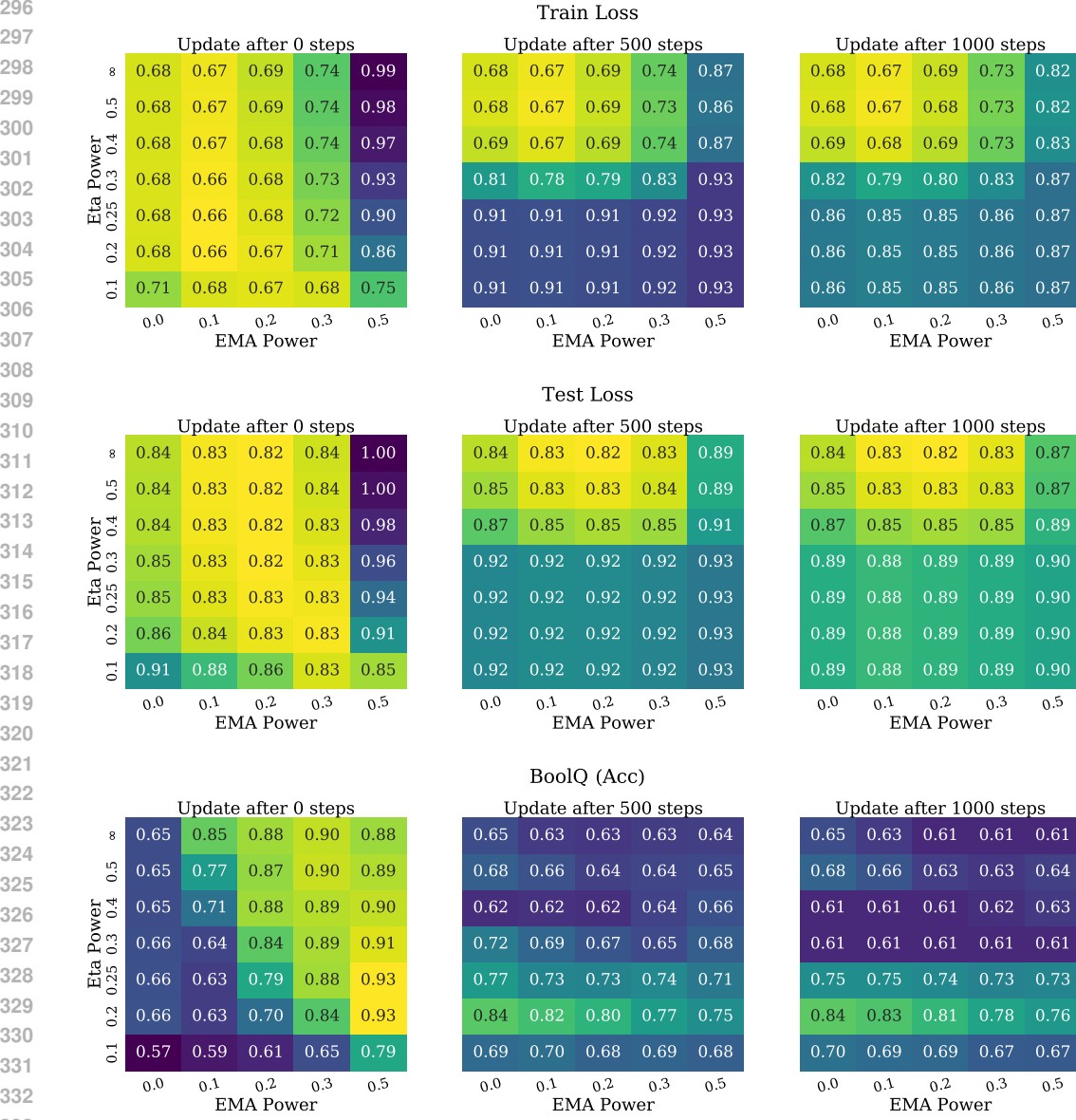

Figure 9: Effect of the choice of $\theta_0$ for different values of $\kappa$ and $\eta$ on optimal throughout training values of train loss (**top**), test loss (**middle**), and BoolQ (**bottom**) for BEMA. In general, choosing $\theta_0$ (**left**) to be the weights of the pre-trained model and immediately applying BEMA leads to the best performance as opposed to waiting 500 (**middle**) or 1000 (**right**) steps before stabilizing. Compared to pure EMA ($\eta = \infty$), which is the top row of each heatmap, BEMA can lead to improved performance.

To establish Kazamaki's criterion (10), we may directly compute that

$$\exp\left(\frac{1}{2}\int_0^T \left\langle \boldsymbol{\Sigma}^{-1}\mathbf{A}(\mu^\star - W_t), dW_t \right\rangle\right) = \exp\left(\frac{1}{2}\left\langle \boldsymbol{\Sigma}^{-1}\mathbf{A}\mu^\star, W_T \right\rangle - \frac{1}{2}\int_0^T \left\langle \boldsymbol{\Sigma}^{-1}\mathbf{A}W_t, dW_t \right\rangle\right).$$

By Ito's rule, it holds that

$$\int_0^T \left\langle \boldsymbol{\Sigma}^{-1}\mathbf{A}W_t, dW_t \right\rangle = \frac{1}{2}\left\langle \boldsymbol{\Sigma}^{-1}\mathbf{A}W_T, W_T \right\rangle - \mathrm{Tr}(\boldsymbol{\Sigma}^{-1}\mathbf{A})T.$$

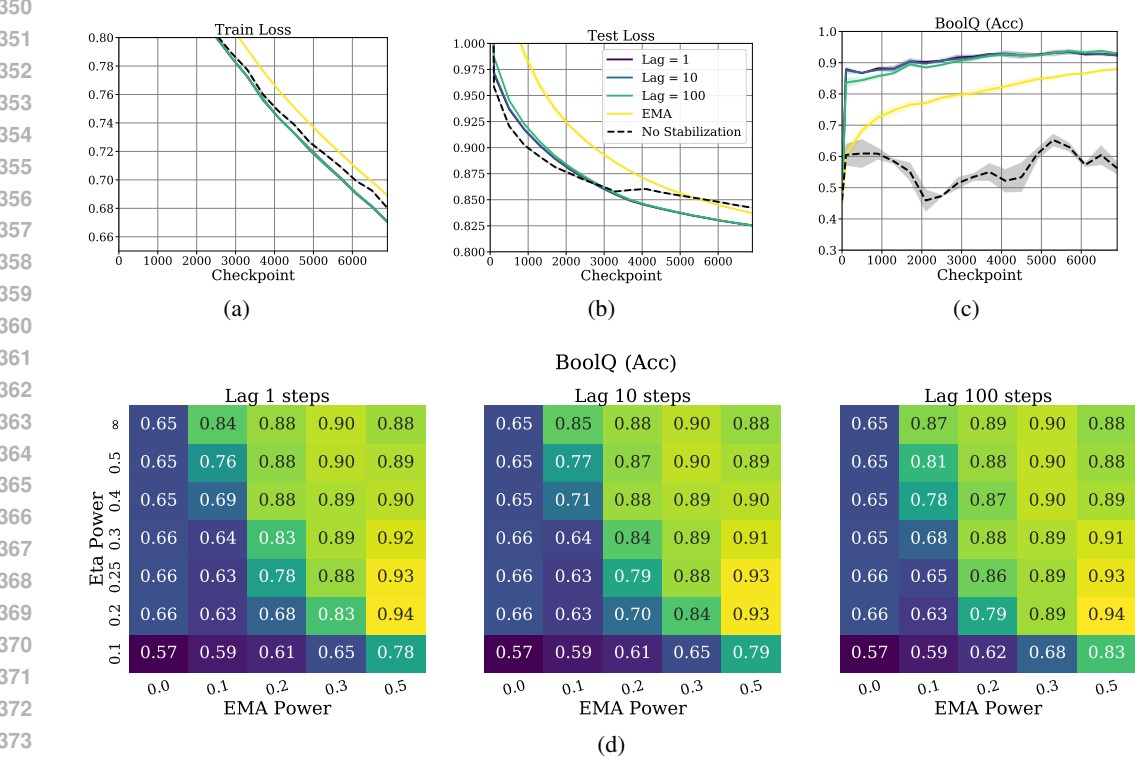

Figure 10: Effect of the choice of lag $\rho$ on training for train loss **(a)**, test loss **(b)**, and BoolQ **(c)**. We also compare the optimal performance throughout training for different values of $\rho$, $\eta$, and $\kappa$ in **(d)**. In general, we see minimal effect of the choice of $\rho$ on performance for all values of $\eta$ and $\kappa$.

As $\mathbf{\Sigma}^{-1}\mathbf{A}$ is positive definite, it then holds that

$$\exp\left(\frac{1}{2}\left\langle\mathbf{\Sigma}^{-1}\mathbf{A}\mu^\star, W_T\right\rangle - \frac{1}{2}\int_0^T\left\langle\mathbf{\Sigma}^{-1}\mathbf{A}W_t, dW_t\right\rangle\right) \leq \exp\left(\frac{1}{2}\left\langle\mathbf{\Sigma}^{-1}\mathbf{A}\mu^\star, W_T\right\rangle + \mathrm{Tr}\left(\mathbf{\Sigma}^{-1}\mathbf{A}\right)T\right).$$

The finiteness of the expectation of this last expression then follows from the fact that $W_T$ is Gaussian and the exponent is an affine function thereof. Thus, (10) holds and the result follows. $\square$

We now recall several useful properties of the OU process. To begin, we recall the standard fact that (2) admits the following closed form solution (see, e.g., Le Gall (2016); Mandt et al. (2015)):

$$\theta_t = e^{-\mathbf{A}t}\theta_0 + \left(\mathbf{I} - e^{-\mathbf{A}t}\right)\mu^\star + \sqrt{\eta}\int_0^t e^{-\mathbf{A}(t-s)}\mathbf{\Sigma}\,dW_s, \tag{11}$$

which we use. Critically, (11) implies that $\theta_t$ is a Gaussian process with mean $\mu_t = e^{-\mathbf{A}t}\theta_0 + \left(\mathbf{I} - e^{-\mathbf{A}t}\right)\mu^\star$ and a simple covariance kernel, given in the following lemma.

**Lemma 1.** *Let $(\theta_t)_{0\leq t\leq T}$ be the solution to the OU process (2) with $\mathbf{A}, \mathbf{\Sigma} \in \mathbb{R}^{d\times d}$ symmetric positive definite. Then, for $0 \leq s < t \leq T$, we have that*

$$\mathrm{Cov}(\theta_t, \theta_s) = K(t,s) = \frac{\eta\cdot\mathbf{A}^{-1}}{2}\int_0^s e^{-\mathbf{A}(t-u)}\mathbf{\Sigma}^2 e^{-\mathbf{A}(s-u)}du \preceq \eta\cdot\|\mathbf{\Sigma}\|_{\mathrm{op}}^2\frac{\mathbf{A}^{-1}}{2}\left(e^{-\mathbf{A}(t-s)} - e^{-\mathbf{A}(t+s)}\right).$$

*Moreover, when $\mathbf{\Sigma} = \sigma\mathbf{I}$, it holds that*

$$\mathrm{Cov}(\theta_t, \theta_s) = \frac{\sigma^2\eta}{2}\mathbf{A}^{-1}\left(e^{-\mathbf{A}|t-s|} - e^{-\mathbf{A}(t+s)}\right).$$

*Proof.* This is a standard fact about OU processes. See, e.g. Le Gall (2016); Kutoyants (2013); Mandt et al. (2015). Indeed, this follows immediately from (11). $\square$

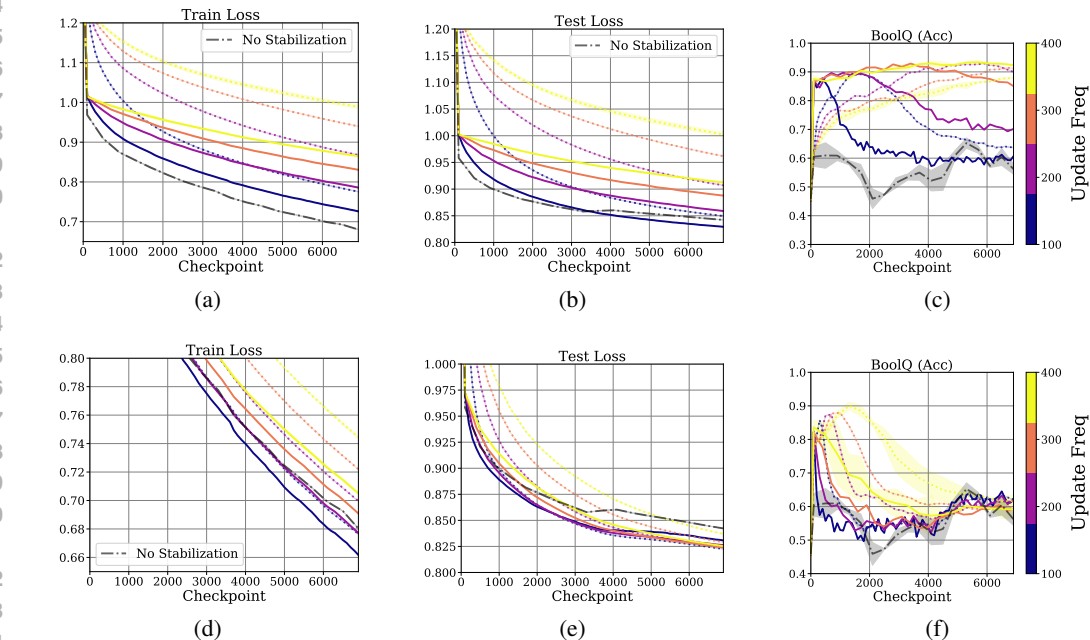

Figure 11: Effect of the chocie of update frequency $\phi$ on train loss **(a,d)**, test loss **(b,e)**, and BoolQ performance **(c,f)** for $\kappa = 0.5$ **(top)** and $\kappa = 0.3$ **(bottom)**. Here we plot both BEMA (solid lines) and EMA (dashed lines) and the color corresponds to $\phi$. Updating with increasing frequency tends to substantially increase convergence speed, leading to significant improvements in train and test losses. This benefit trades off against (a) compute time in that more frequent updates slow the wall clock time of training and (b) potential overfitting, as can be observed in the BoolQ performance for $\phi = 100$ and, to a lesser extent, $\phi = 200$.

We now require three lemmata that handle the first and second order moments of transformations of the OU process we use throughout the paper. The first controls the first two moments of the total displacement of the OU process.

**Lemma 2.** *Let $\theta_t$ denote the solution to* (2) *given by* (11)*. Then it holds that*

$$\mathbb{E}\left[\theta_T - \theta_0\right] = \left(\mathbf{I} - e^{-\mathbf{A}T}\right)\left(\mu^\star - \theta_0\right) \qquad and \qquad \mathrm{Cov}\left(\theta_T - \theta_0\right) \preceq \eta \cdot \|\mathbf{\Sigma}\|_{\mathrm{op}}^2 \cdot \mathbf{A}^{-1}\left(\mathbf{I} - e^{-2\mathbf{A}T}\right)$$

*with equality in the variance when $\mathbf{\Sigma} = \sigma\mathbf{I}$.*

*Proof.* By (11), it holds that

$$\theta_t = e^{-\mathbf{A}t}\theta_0 + \left(\mathbf{I} - e^{-\mathbf{A}t}\right)\mu^\star + \sqrt{\eta}\int_0^t e^{-\mathbf{A}(t-s)}\mathbf{\Sigma}\,dW_s.$$

Note that the expectation of the final term is zero because this is a martingale. The first equality then follows immediately. For the variance, we observe that because $\theta_0$ is deterministic, it holds by Lemma 1 that

$$\mathrm{Cov}(\theta_T - \theta_0) = \mathrm{Cov}(\theta_T) \preceq \eta\|\mathbf{\Sigma}\|_{\mathrm{op}}^2 \cdot \mathbf{A}^{-1}\left(\mathbf{I} - e^{-2\mathbf{A}T}\right),$$

with equality in the case that $\mathbf{\Sigma} = \sigma\mathbf{I}$. The result follows. $\qquad\square$

We now require an analogous result for the time average of a trajectory of the OU process.

**Lemma 3.** *Let $\theta_t$ be the solution to* (2) *given by* (11)*. Then it holds that*

$$\mathbb{E}\left[\frac{1}{T}\int_0^T \theta_t\,dt\right] = \mu^\star - \frac{1}{T}\mathbf{A}^{-1}\left(\mathbf{I} - e^{-\mathbf{A}T}\right)\left(\mu^\star - \theta_0\right).$$

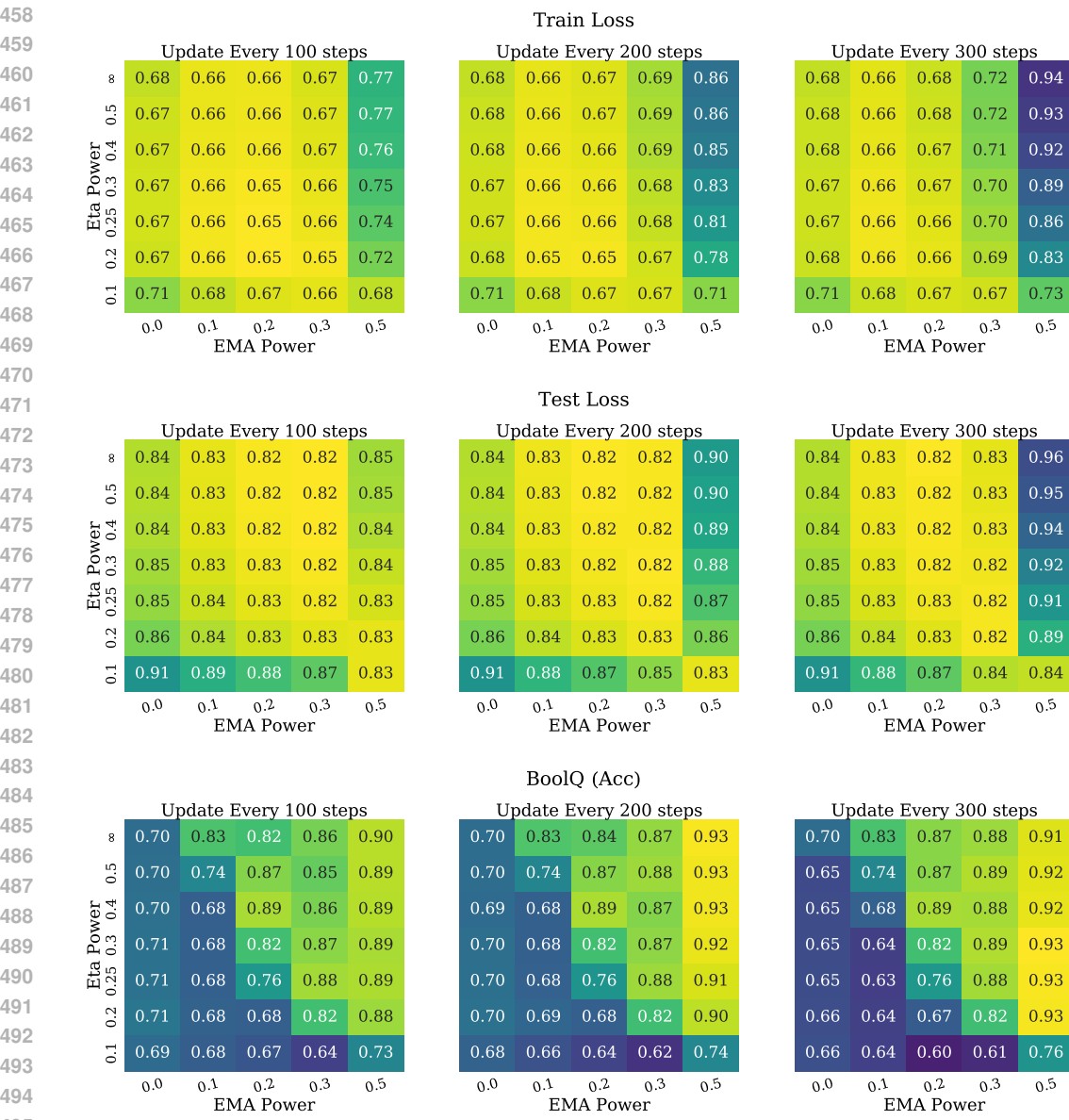

Figure 12: Effect of the choice of update frequency $\phi$ on optimal throughout training trajectory crossentropy loss on train **(top)** and test **(middle)** sets as well as performance on BoolQ **(bottom)** for a variety of choices of $\kappa$ and $\eta$. Compare to Figures 7 and 8 for the default choice of $\phi = 400$.

*Moreover,*

$$\eta \lambda_{\min}(\Sigma)^2 \mathbf{A}^{-2} \left( T \cdot \mathbf{I} - \mathbf{A}^{-1} \left[ 2 \left( \mathbf{I} - e^{-\mathbf{A}T} \right) - \frac{1}{2} \left( \mathbf{I} - e^{-2\mathbf{A}T} \right) \right] \right) \preceq \mathrm{Cov} \left( \int_0^T \theta_t \, dt \right)$$
$$\preceq T \cdot \eta \left\| \mathbf{\Sigma} \right\|_{\mathrm{op}}^2 \cdot \mathbf{A}^{-2}.$$

*In the case that $\mathbf{\Sigma} = \sigma \mathbf{I}$, the first inequality above is an equality.*

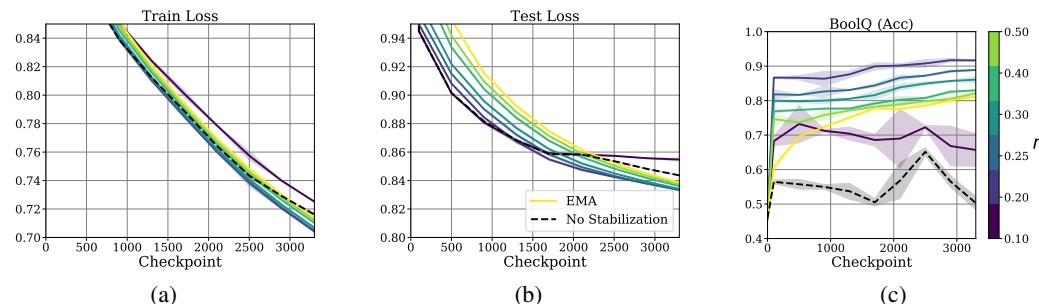

(a)   (b)   (c)

Figure 13: Demonstration of the robustness of BEMA performance improvements to the choice of batch size; here training is conducted with an effective batch size of 512 and train loss **(a)**, test losses **(b)**, and BoolQ performance **(c)** are shown. We continue to see considerable performance improvements over EMA.

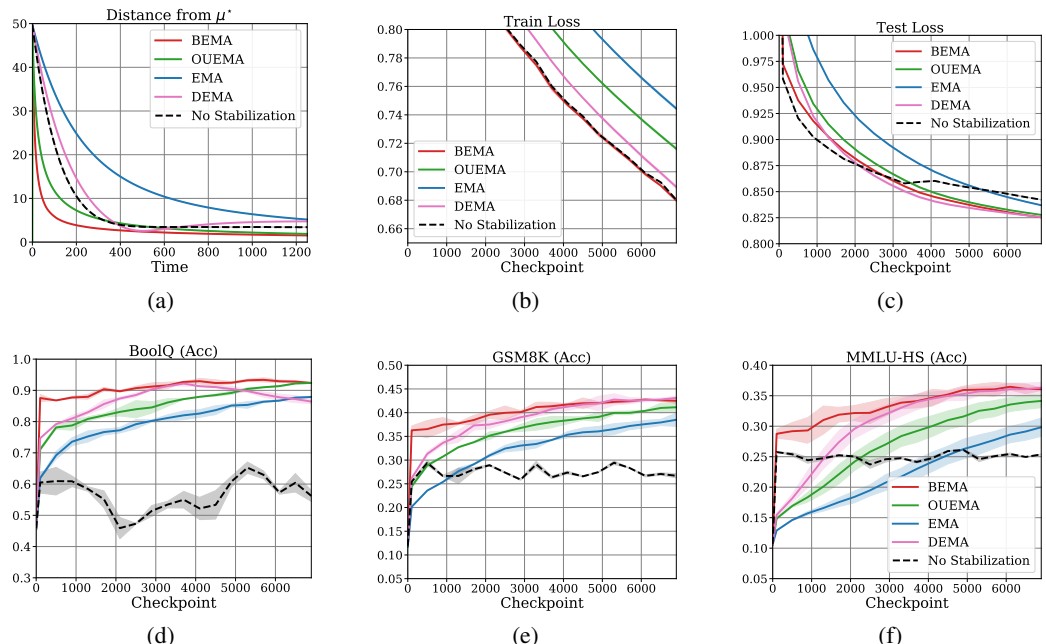

Figure 14: Comparison of BEMA to alternative stabilizer DEMA. **(a)** Demonstration of the effect of DEMA on a quadratic loss landscape. **(b)** Train loss, **(c)** test loss, **(d)** BoolQ performance, **(e)** GSM8K performance, and **(f)** MMLU-HS performance for DEMA compared to BEMA, OUEMA, and EMA. In general, DEMA improves on OUEMA, which improves on EMA, but neither matches the acceleration and performance of BEMA on the generation benchmarks.

*Proof.* For the first statement, note that by the lineary of expectation it holds that

$$\mathbb{E}\left[\frac{1}{T}\int_0^T \theta_t dt\right] = \frac{1}{T}\int_0^T \mathbb{E}\left[\theta_t\right]dt$$

$$= \mu^\star - \frac{1}{T}\int_0^T e^{-\mathbf{A}t}\left(\mu^\star - \theta_0\right)dt$$

$$= \mu^\star - \frac{1}{T}\mathbf{A}^{-1}\left(\mathbf{I} - e^{-\mathbf{A}T}\right)\left(\mu^\star - \theta_0\right).$$

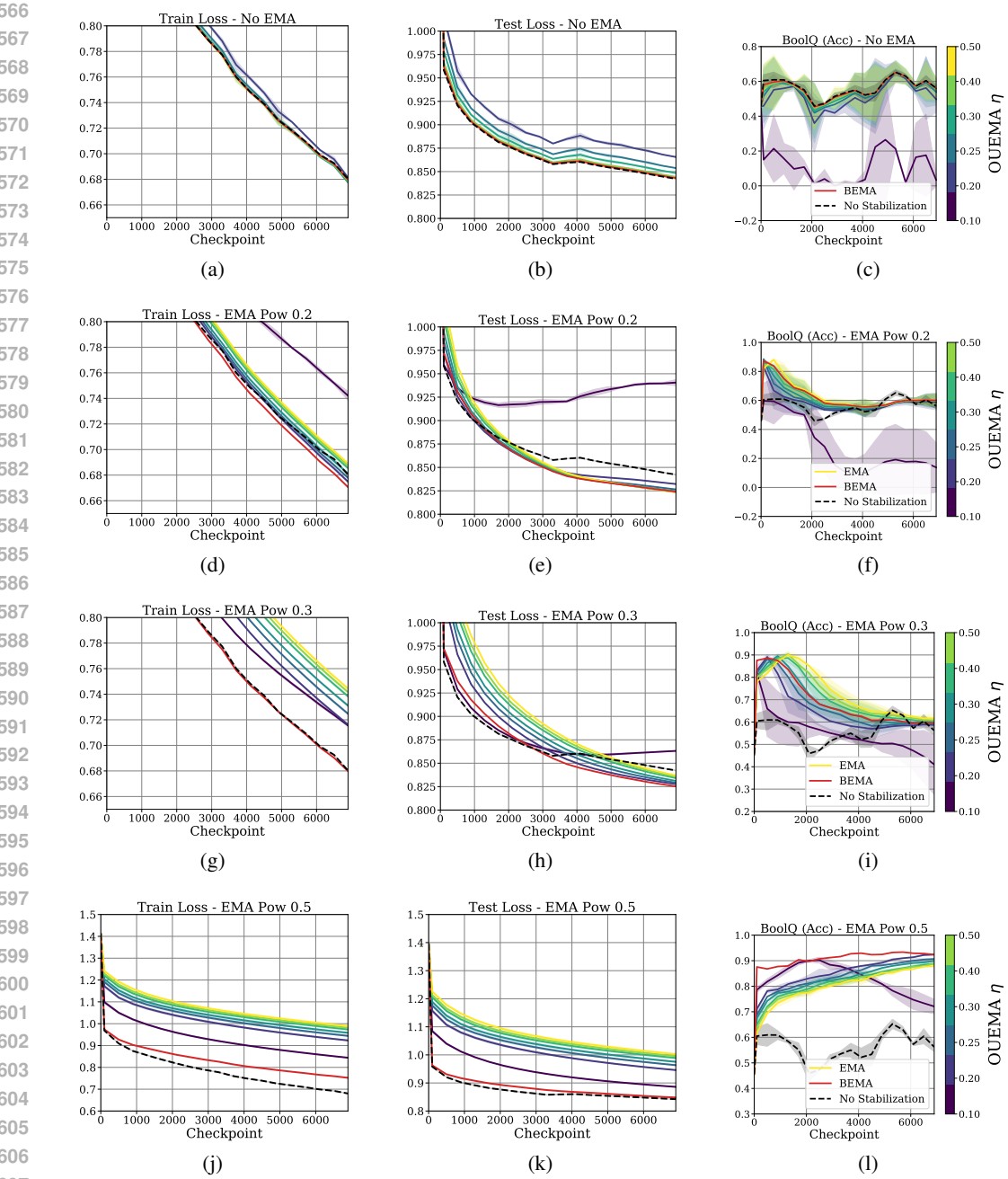

Figure 15: Effect of OUEMA for different values of $\kappa$ from $\kappa = 0.0$ (no EMA **top**) to $\kappa = 0.5$ (strongest EMA **bottom**). We compare to vanilla optimization (No stabilization, dashed), EMA (yellow), and BEMA for the best choice of $\eta$ (red). BEMA is generally superior to OUEMA and EMA.

For the covaraiance, we apply Lemma 1 and see that by symmetry

$$\text{Cov}\left(\int_0^T \theta_t \, dt\right) = \int_0^T \int_0^T \text{Cov}(\theta_t, \theta_u) du dt = 2 \int_0^T \int_0^T K(t, u) du dt,$$

where $K(t, u)$ is as in Lemma 1. We have that

$$\eta \lambda_{\min}(\boldsymbol{\Sigma})^2 \frac{\mathbf{A}^{-1}}{2}\left(e^{-\mathbf{A}|t-s|} - e^{-\mathbf{A}(t+s)}\right) \preceq K(s, t) \preceq \eta \lambda_{\max}(\boldsymbol{\Sigma})^2 \frac{\mathbf{A}^{-1}}{2}\left(e^{-\mathbf{A}|t-s|} - e^{-\mathbf{A}(t+s)}\right).$$

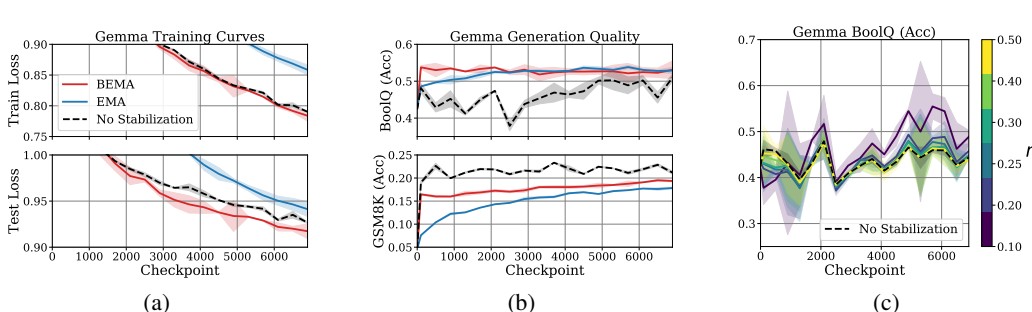

(a)        (b)        (c)

Figure 16: Performance of BEMA and EMA on Gemma3-1B for **(a)** train and test loss, **(b)** generations on BoolQ **(top)** and GSM8K **(bottom)**. We also show the effect of BEMA with $\kappa = 0$ (no EMA) for a variety of choices of $\eta$ in **(c)**. In general, BEMA accelerates and improves on EMA performance, but the effect is less pronounced than for Qwen2.5-1.5B.

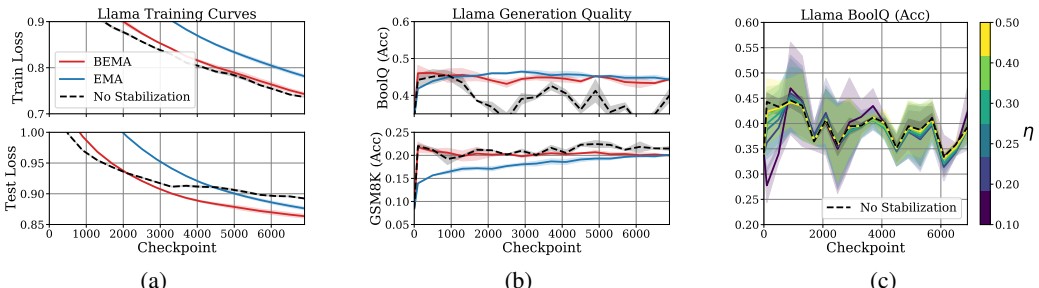

(a)        (b)        (c)

Figure 17: Performance of BEMA and EMA on Llama3.2-1B for **(a)** train and test loss, **(b)** generations on BoolQ **(top)** and GSM8K **(bottom)**. We also show the effect of BEMA with $\kappa = 0$ (no EMA) for a variety of choices of $\eta$ in **(c)**. It is clear that BEMA is an improvement with respect to train and test loss, but Llama3.2-1B does not follow commands with sufficient frequency so as to perform sufficiently in either GSM8K or BoolQ after finetuning on Tulu-3-SFT in order to recover a clear signal.

Moreover, we compute

$$\int_0^T \int_0^t \mathbf{A}^{-1} \left( e^{-\mathbf{A}|t-s|} - e^{-\mathbf{A}(t+s)} \right) \, ds dt = \mathbf{A}^{-2} \int_0^T \left( \mathbf{I} - 2e^{-\mathbf{A}t} + e^{-2\mathbf{A}t} \right) \, dt$$

$$= \mathbf{A}^{-2} \left( T \cdot \mathbf{I} - \mathbf{A}^{-1} \left[ 2 \left( \mathbf{I} - e^{-\mathbf{A}T} \right) - \frac{1}{2} \left( \mathbf{I} - e^{-2\mathbf{A}T} \right) \right] \right).$$

Plugging this into the above display yileds the left hand side inequality, as well as the equality when $\lambda_{\min}(\mathbf{\Sigma}) = \lambda_{\max}(\mathbf{\Sigma})$. For the upper bound, we see that by the positive definiteness of $\mathbf{A}$, we may diagonalize $\mathbf{A}$ and it suffices to demonstrate that for any $x \geq 0$, it holds that

$$2(1 - e^{-x}) - \frac{1 - e^{-2x}}{2} \geq 0.$$

Letting $u = e^{-x}$, we see that this is equivalent to showing that $u^2 - 4u + 3 \geq 0$ when $0 \leq u \leq 1$, which is immediate. The result follows. $\square$

Finally, we require control on the covariance between the total displacement and the time average of the OU process.

**Lemma 4.** *Let $\theta_t$ be the solution to (2) given by (11). Then it holds that*

$$\eta \lambda_{\min}(\mathbf{\Sigma})^2 \frac{\mathbf{A}^{-2}}{2} \left( \mathbf{I} - 2e^{-\mathbf{A}T} + e^{-2\mathbf{A}T} \right) \preceq \mathrm{Cov} \left( \theta_T - \theta_0, \int_0^T \theta_t dt \right) \preceq \eta \|\mathbf{\Sigma}\|_{\mathrm{op}}^2 \frac{\mathbf{A}^{-2}}{2},$$

*with equality on the left hand side $\mathbf{\Sigma}$ is a scalar multiple of the identity.*

*Proof.* By linearity of expectation and the fact that $\theta_0$ is deterministic, it holds that

$$\mathrm{Cov} \left( \theta_T - \theta_0, \int_0^T \theta_t \, dt \right) = \mathrm{Cov} \left( \theta_T, \int_0^T \theta_t \, dt \right)$$

$$= \int_0^T \mathrm{Cov} \left( \theta_T, \theta_t \right) \, dt$$

$$= \int_0^T K(T, t) \, dt,$$

where $K(t, s)$ is as in Lemma 1. Using the bounds on $K(t, s)$ from Lemma 1, we see that

$$\int_0^T \frac{\mathbf{A}^{-1}}{2} \left( e^{-\mathbf{A}|T-t|} - e^{-\mathbf{A}(T+t)} \right) \, dt = \frac{\mathbf{A}^{-2}}{2} \left( \mathbf{I} - e^{-\mathbf{A}T} + e^{-2\mathbf{a}T} \right).$$

The result follows. $\square$

Finally, we precisely characterize the error of $\theta_T$ as an estimator of $\mu^\star$.

**Proposition 5.** *For $T > 0$, let $(\theta_t)_{0 \leq t \leq T}$ be the solution to the OU process (2) with $\mathbf{A}, \mathbf{\Sigma} \in \mathbb{R}^{d \times d}$ symmetric positive definite. Then it holds that*

$$\mathbb{E}_{\mu^\star} \left[ \|\theta_T - \mu^\star\|^2 \right] \leq \left\| e^{-\mathbf{A}T} \left( \theta_0 - \mu^\star \right) \right\|^2 + \eta \cdot \|\mathbf{\Sigma}\|_{\mathrm{op}}^2 \cdot \mathrm{Tr} \left( \mathbf{A}^{-1} \right).$$

*If $\mathbf{\Sigma} = \sigma \mathbf{I}$, then the inequality becomes an equality.*

*Proof.* By the bias-variance decomposition, it holds that

$$\mathbb{E}_{\mu^\star} \left[ \|\theta_T - \mu^\star\|^2 \right] = \|\mathbb{E}_{\mu^\star} [\theta_T] - \mu^\star\|^2 + \mathrm{Tr} \left( \mathrm{Cov}(\theta_T) \right).$$

Applying Lemma 2 concludes the proof. $\square$

### E.2 Lower Bound on Mean Squared Error

We now state and prove two lower bounds on the mean squared error of estimators of $\mu^\star$ based on the OU process. Both bounds are a consequence of the Cramer-Rao inequality, the main approach in classical statistics to derive lower bounds in parametric estimation problems. We first prove a result for unbiased estimators, which is a consequence of the Cramer-Rao inequality.

**Proposition 6.** *Let $(\theta_t)_{0 \le t \le T}$ be the solution to the OU process (2) with $\mathbf{A}, \mathbf{\Sigma} \in \mathbb{R}^{d \times d}$ symmetric positive definite and let $\widehat{\mu}$ be an unbiased estimator of $\mu^\star$, i.e., $\mathbb{E}_{\mu^\star}[\widehat{\mu}] = \mu^\star$. Then it holds that*

$$\mathbb{E}\left[\|\widehat{\mu} - \mu^\star\|^2\right] \ge \frac{\eta \cdot \mathrm{Tr}\left(\mathbf{A}^{-1}\mathbf{\Sigma}^2\mathbf{A}^{-1}\right)}{T}.$$

*In particular, if $\mathbf{\Sigma} = \sigma\mathbf{I}$, then it holds that*

$$\mathbb{E}\left[\|\widehat{\mu} - \mu^\star\|^2\right] \ge \frac{\sigma^2\eta}{T} \cdot \mathrm{Tr}\left(\mathbf{A}^{-2}\right).$$

*Proof.* We apply the Cramer-Rao inequality to diffusions, as in Liptser & Shiryaev (2013a;b); Kutoyants (2013). Indeed, by the Cramer-Rao inequality in multiple dimensions (see, e.g., Liptser & Shiryaev (2013a, §7.8) or Lehmann & Casella (2006, Theorem 6.1)) it holds that

$$\mathbb{E}_{\mu^\star}\left[(\widehat{\mu} - \mathbb{E}_{\mu^\star}[\widehat{\mu}])(\widehat{\mu} - \mathbb{E}_{\mu^\star}[\widehat{\mu}])^\top\right] \succeq (\nabla_{\mu^\star}\mathbb{E}_{\mu^\star}[\widehat{\mu}]) \, \mathbb{I}_{\mu^\star}^{-1} \, (\nabla_{\mu^\star}\mathbb{E}_{\mu^\star}[\widehat{\mu}])^\top, \qquad (12)$$

where

$$\mathbb{I}_{\mu^\star} = \mathbb{E}_{\mu^\star}\left[-\nabla^2 \log p_{\mu^\star}(\theta)\right]$$

is the Fisher information matrix of the process $\mathbb{P}^\mu$ with respect to the parameter $\mu^\star$ and $\nabla_{\mu^\star}\mathbb{E}_{\mu^\star}[\widehat{\mu}]$ is the Jacobian of the expectation of $\widehat{\mu}$ with respect to $\mu^\star$. In the case that $\widehat{\mu}$ is unbiased, we have that $\nabla_{\mu^\star}\mathbb{E}_{\mu^\star}[\widehat{\mu}] = \mathbf{I}$, and thus the Cramer-Rao inequality tells us that $\mathrm{Cov}(\widehat{\mu}) \succeq \mathbb{I}_{\mu^\star}^{-1}$. By the bias-variance decomposition, it holds that

$$\mathbb{E}_{\mu^\star}\left[\|\widehat{\mu} - \mu^\star\|^2\right] = \|\mathbb{E}_{\mu^\star}[\widehat{\mu}] - \mu^\star\|^2 + \mathrm{Tr}\left(\mathrm{Cov}(\widehat{\mu})\right).$$

In the case that $\widehat{\mu}$ is unbiased, then, we have that $\mathbb{E}_{\mu^\star}\left[\|\widehat{\mu} - \mu^\star\|^2\right] \ge \mathrm{Tr}\left(\mathbb{I}_{\mu^\star}^{-1}\right)$. We now use Theorem 3 to compute the Fisher information matrix for the OU process. Indeed, we have by Proposition 4 that

$$\log p_{\mu^\star}(\theta_t) = -\eta^{-1/2}\int_0^T \left\langle \mathbf{\Sigma}^{-1}\mathbf{A}(\mu^\star - \theta_t), d\theta_t\right\rangle - \frac{1}{2\eta}\int_0^T \left\|\mathbf{\Sigma}^{-1}\mathbf{A}(\mu^\star - \theta_t)\right\|^2 dt$$

Taking the Hessian with respect to $\mu^\star$ yields

$$-\nabla^2 \log p_{\mu^\star}(\theta_t) = \eta^{-1}\int_0^T \mathbf{A}\mathbf{\Sigma}^{-2}\mathbf{A}\, dt = \eta^{-1}T\mathbf{A}\mathbf{\Sigma}^{-2}\mathbf{A}.$$

The result follows. $\qquad\square$

In addition Proposition 6, which only holds for unbiased estimators, we also have a lower bound that holds when the bias is a *contraction*, i.e., is Lipschitz with parameter $L < 1$. This also follows from the Cramer-Rao inequality.

**Proposition 7.** *Let $(\theta_t)_{0 \le t \le T}$ be the solution to the OU process (2) with $\mathbf{A}, \mathbf{\Sigma} \in \mathbb{R}^{d \times d}$ symmetric positive definite and let $\widehat{\mu}$ be an estimator of $\mu^\star$ such that the map $\mu^\star \mapsto \mathbb{E}_{\mu^\star}[\widehat{\mu}] - \mu^\star$ is Lipschitz with constant $L < 1$. Then it holds that*

$$\mathbb{E}\left[\|\widehat{\mu} - \mu^\star\|^2\right] \ge (1 - L)^2 \cdot \frac{\eta \cdot \mathrm{Tr}\left(\mathbf{A}^{-1}\mathbf{\Sigma}^2\mathbf{A}^{-1}\right)}{T} \ge (1 - L)^2 \cdot \frac{\eta\lambda_{\min}(\mathbf{\Sigma})^2 \cdot \mathrm{Tr}\left(\mathbf{A}^{-2}\right)}{T}.$$

*Proof.* We may apply the identical argument as in the proof of Proposition 6, i.e., following from (12) we have

$$\mathbb{E}_{\mu^\star}\left[\|\widehat{\mu} - \mu^\star\|^2\right] \ge \frac{\eta}{T} \cdot \mathrm{Tr}\left((\nabla_{\mu^\star}\mathbb{E}_{\mu^\star}[\widehat{\mu}])^\top \mathbf{A}^{-1}\mathbf{\Sigma}^2\mathbf{A}^{-1}(\nabla_{\mu^\star}\mathbb{E}_{\mu^\star}[\widehat{\mu}])\right).$$

By the linearity of the Jacobian, it holds that

$$\nabla_{\mu^\star}\mathbb{E}_{\mu^\star}[\widehat{\mu}] = \nabla_{\mu^\star}\left(\mathbb{E}_{\mu^\star}[\widehat{\mu}] - \mu^\star\right) + \nabla_{\mu^\star}\mu^\star \succeq \left(1 - \left\|\nabla_{\mu^\star}\left(\mathbb{E}_{\mu^\star}[\widehat{\mu}] - \mu^\star\right)\right\|_{\mathrm{op}}\right)\mathbf{I}.$$

By the Lipschitz condition, we have that $\left\|\nabla_{\mu^\star}\left(\mathbb{E}_{\mu^\star}[\widehat{\mu}] - \mu^\star\right)\right\|_{\mathrm{op}} \le L < 1$ and thus the result follows. $\qquad\square$

### E.3 Maximum Likelihood Estimation

We now prove several results related to the Maximum Likelihood Estimator (MLE) $\widehat{\mu}^{\mathsf{MLE}}$ of the OU process. We begin by providing an explicit formula for the MLE, which is an immediate consequence of Proposition 4 and the definition of the MLE. Such a formula, especially in one dimension, is well-known in the financial mathematics literature (Kutoyants, 2013), but we include it here for completeness.

**Theorem 4.** *For any $T > 0$, let $(\theta_t)_{0 \leq t \leq T}$ be the solution to the OU process (2) with $\mathbf{A}, \mathbf{\Sigma} \in \mathbb{R}^{d \times d}$ symmetric positive definite. Then the Maximum Likelihood Estimator (MLE) of $\mu^\star$ is given by*

$$\widehat{\mu}_T^{\mathsf{MLE}} = \frac{\mathbf{A}^{-1}}{T} (\theta_T - \theta_0) + \frac{1}{T} \int_0^T \theta_t \, dt.$$

*Proof.* We have by Proposition 4 that the log likelihood function is given by

$$\eta \cdot L(\mu) = \eta \cdot \log \frac{d\mathbb{P}^\mu}{d\mathbb{P}^W} = - \int_0^T \left\langle \mathbf{\Sigma}^{-2} \mathbf{A} \left( \mu - \theta_t \right), d\theta_t \right\rangle - \frac{1}{2} \int_0^T \left\| \mathbf{\Sigma}^{-1} \mathbf{A} \left( \mu - \theta_t \right) \right\|^2 dt.$$

Note that this function is strongly concave in $\mu$ and thus attains a unique maximum at the stationary point where $\nabla L(\widehat{\mu}_T^{\mathsf{MLE}}) = 0$. Taking the gradient, we see that

$$0 = \nabla L(\widehat{\mu}_T^{\mathsf{MLE}}) = \int_0^T \left\langle \mathbf{\Sigma}^{-2} \mathbf{A}, d\theta_t \right\rangle - \int_0^t \mathbf{A} \mathbf{\Sigma}^{-2} \mathbf{A} \left( \widehat{\mu}_T^{\mathsf{MLE}} - \theta_t \right) dt$$

$$= \mathbf{A} \mathbf{\Sigma}^{-2} (\theta_T - \theta_0) - \mathbf{A} \mathbf{\Sigma}^{-2} \mathbf{A} \left( T \cdot \widehat{\mu}_T^{\mathsf{MLE}} - \int_0^T \theta_t dt \right).$$

Rearranging yields the desires conclusion. $\square$

We now use this characterization of the MLE to derive its distributional properties.

**Corollary 1.** *Let $(\theta_t)_{0 \leq t \leq T}$ be the solution to the OU process (2) with $\mathbf{A}, \mathbf{\Sigma} \in \mathbb{R}^{d \times d}$ symmetric positive definite. Then it holds that*

$$\widehat{\mu}_T^{\mathsf{MLE}} \stackrel{d}{=} \mu^\star + \frac{\sqrt{\eta} \cdot \mathbf{A}^{-1} \mathbf{\Sigma}}{\sqrt{T}} \cdot \mathcal{N} (0, \mathbf{I}),$$

*where $\stackrel{d}{=}$ denotes equality in distribution. In particular, it holds that*

$$\mathbb{E} \left[ \left\| \widehat{\mu}_T^{\mathsf{MLE}} - \mu^\star \right\|^2 \right] = \frac{\eta \cdot \mathrm{Tr} \left( \mathbf{A}^{-1} \mathbf{\Sigma}^2 \mathbf{A}^{-1} \right)}{T}$$

*and, in the special case that $\mathbf{\Sigma} = \sigma \mathbf{I}$, it holds that*

$$\mathbb{E} \left[ \left\| \widehat{\mu}_T^{\mathsf{MLE}} - \mu^\star \right\|^2 \right] = \frac{\sigma^2 \eta \cdot \mathrm{Tr} \left( \mathbf{A}^{-2} \right)}{T}.$$

*Proof.* By Theorem 4 it holds that

$$\widehat{\mu}_T^{\mathsf{MLE}} = \frac{\mathbf{A}^{-1}}{T} (\theta_T - \theta_0) + \frac{1}{T} \int_0^T \theta_t \, dt$$

$$= \frac{1}{T} \left( \mathbf{A}^{-1} \int_0^T \mathbf{A} (\mu^\star - \theta_t) \, dt + \mathbf{A}^{-1} m \int_0^T \mathbf{\Sigma} dW_t + \int_0^T \theta_t dt \right)$$

$$= \mu^\star + \frac{\mathbf{A}^{-1} \mathbf{\Sigma} W_T}{T}.$$

The result now follows from the fact that $W_T \sim \mathcal{N}(0, T\mathbf{I})$. $\square$

We now prove a general result on the performance of estimates of the form $\widehat{\mu}^{\mathsf{MLE}}$, but with a possibly different choice of $\mathbf{A}$. Let

$$\widetilde{\mu}_T^{\mathsf{MLE}}(\widetilde{\mathbf{A}}) = \frac{\mathbf{A}^{-1}}{T} \left( \theta_T - \theta_0 \right) + \frac{1}{T} \int_0^T \theta_t \, dt,$$

where $\widetilde{\mathbf{A}}$ is a symmetric positive definite matrix. We have the following bound on the performance of such an estimator.

**Theorem 5.** *Let* $(\theta_t)_{0 \leq t \leq T}$ *be the solution to the OU process* (2) *with* $\mathbf{A}, \boldsymbol{\Sigma} \in \mathbb{R}^{d \times d}$ *symmetric positive definite. Then it holds that*

$$\mathbb{E}\left[ \left\| \widetilde{\mu}_T^{\mathsf{MLE}}(\widetilde{\mathbf{A}}) - \mu^\star \right\|^2 \right] \leq \frac{\eta \left\| \boldsymbol{\Sigma} \right\|_{\mathrm{op}}^2 \cdot \mathrm{Tr}\left( \mathbf{A}^{-2} \right)}{T} + \frac{\eta \left\| \boldsymbol{\Sigma} \right\|_{\mathrm{op}}^2 \cdot \left\| \widetilde{\mathbf{A}}^{-1} \right\|_{\mathrm{op}}^2 \cdot \mathrm{Tr}\left( \mathbf{A}^{-1} \right)}{T^2}$$

$$+ \frac{\eta \left\| \boldsymbol{\Sigma} \right\|_{\mathrm{op}}^2 \left\| \widetilde{\mathbf{A}}^{-1} \right\|_{\mathrm{op}} \mathrm{Tr}\left( \mathbf{A}^{-2} \right)}{T^2} + \frac{\left\| \widetilde{\mathbf{A}}^{-1} - \mathbf{A}^{-1} \right\|_{\mathrm{op}}^2 \left\| \mu^\star - \theta_0 \right\|^2}{T^2}$$

*If* $\boldsymbol{\Sigma} = \sigma \mathbf{I}$, *and* $\widetilde{\mathbf{A}}$ *commutes with* $\mathbf{A}$, *then it holds that*

$$\mathbb{E}\left[ \left\| \widetilde{\mu}_T^{\mathsf{MLE}}(\widetilde{\mathbf{A}}) - \mu^\star \right\|^2 \right] \leq \frac{\eta \sigma^2 \, \mathrm{Tr}\left( \mathbf{A}^{-2} \right)}{T} + \frac{\eta \sigma^2 \, \mathrm{Tr}\left( \widetilde{\mathbf{A}}^{-2} \mathbf{A}^{-1} \right)}{T^2} + \frac{\eta \sigma^2 \, \mathrm{Tr}\left( \widetilde{\mathbf{A}}^{-1} \mathbf{A}^{-2} \right)}{T^2} \quad (13)$$

$$+ \frac{\left\| \widetilde{\mathbf{A}}^{-1} - \mathbf{A}^{-1} \right\|_{\mathrm{op}}^2 \left\| \mu^\star - \theta_0 \right\|^2}{T^2}.$$

*Proof.* We apply the bias-variance decomposition and bound each separately. For the bias, we combine the first moment bounds of Lemmas 2 and 3 and the linearity of expectation to see that

$$\mathbb{E}\left[ \widetilde{\mu}_T^{\mathsf{MLE}}(\widetilde{\mathbf{A}}) - \mu^\star \right] = \frac{\widetilde{\mathbf{A}}^{-1}}{T} \left( \mathbf{I} - e^{-\mathbf{A}T} \right) (\mu^\star - \theta_0) - \frac{1}{T} \mathbf{A}^{-1} \left( \mathbf{I} - e^{-\mathbf{A}T} \right) (\mu^\star - \theta_0)$$

$$= \frac{\widetilde{\mathbf{A}}^{-1} - \mathbf{A}^{-1}}{T} \left( \mathbf{I} - e^{-\mathbf{A}T} \right) (\mu^\star - \theta_0).$$

For the variance, we apply Lemmas 2 to 4 to see that

$$\mathrm{Var}(\widetilde{\mu}_T^{\mathsf{MLE}}(\widetilde{\mathbf{A}})) = \mathrm{Var}\left( \frac{\widetilde{\mathbf{A}}^{-1}}{T} \theta_T \right) + \mathrm{Var}\left( \frac{1}{T} \int_0^T \theta_t \, dt \right)$$

$$+ \frac{2}{T^2} \cdot \mathrm{Cov}\left( \widetilde{\mathbf{A}}^{-1} \theta_T, \int_0^T \theta_t \, dt \right)$$

$$\leq \frac{\eta \left\| \boldsymbol{\Sigma} \right\|_{\mathrm{op}}^2 \cdot \mathrm{Tr}\left( \mathbf{A}^{-2} \right)}{T} + \frac{\eta \left\| \boldsymbol{\Sigma} \right\|_{\mathrm{op}}^2 \cdot \left\| \widetilde{\mathbf{A}}^{-1} \right\|_{\mathrm{op}}^2 \cdot \mathrm{Tr}\left( \mathbf{A}^{-1} \right)}{T^2} + \frac{\eta \left\| \boldsymbol{\Sigma} \right\|_{\mathrm{op}}^2 \left\| \widetilde{\mathbf{A}}^{-1} \right\|_{\mathrm{op}} \mathrm{Tr}\left( \mathbf{A}^{-2} \right)}{T^2}.$$

The first result follows. For the second result, we can simplify the variance expression using the fact that $\boldsymbol{\Sigma}$ commutes with $\mathbf{A}$ and $\widetilde{\mathbf{A}}$. We have that

$$\mathrm{Var}(\widetilde{\mu}_T^{\mathsf{MLE}}(\widetilde{\mathbf{A}})) = \frac{1}{T^2} \cdot \mathrm{Var}\left( \widetilde{\mathbf{A}}^{-1} \theta_T \right) + \mathrm{Var}\left( \frac{1}{T} \int_0^T \theta_t \, dt \right)$$

$$+ \frac{2}{T^2} \cdot \mathrm{Cov}\left( \widetilde{\mathbf{A}}^{-1} \theta_T, \int_0^T \theta_t \, dt \right)$$

$$\leq \frac{\eta \sigma^2 \, \mathrm{Tr}\left( \widetilde{\mathbf{A}}^{-2} \mathbf{A}^{-1} \right)}{T^2} + \frac{\eta \sigma^2 \, \mathrm{Tr}\left( \mathbf{A}^{-2} \right)}{T} + \frac{\eta \sigma^2 \, \mathrm{Tr}\left( \widetilde{\mathbf{A}}^{-1} \mathbf{A}^{-2} \right)}{T^2}.$$

The second result follows. $\qquad \square$

We see that with respect to asymptotic in $T$ performance, the choice of $\widetilde{\mathbf{A}}$ is irrelevant, as it does not affect the leading term in the error bound. On the other hand, the higher order terms of (13) suggest that $\widetilde{\mathbf{A}}$, subject to being maximally close to $\mathbf{A}$, should be chosen so as to be sufficiently well conditioned in order to temper the additional variance (second term of (13)).

We now instantiate Theorem 5 in order to recover a bound on $\widehat{\mu}_T^{\mathsf{EMA}}$.

**Corollary 2.** *Let $(\theta_t)_{0 \leq t \leq T}$ be the solution to the OU process (2) with $\mathbf{A}, \mathbf{\Sigma} \in \mathbb{R}^{d \times d}$ symmetric positive definite and recall that*

$$\widehat{\mu}_T^{\mathsf{EMA}} = \frac{1}{T} \int_0^T \theta_t \, dt.$$

*Then*

$$\mathbb{E}\left[\left\|\widehat{\mu}_T^{\mathsf{EMA}} - \mu^\star\right\|^2\right] \leq \frac{\eta \left\|\mathbf{\Sigma}\right\|_{\mathrm{op}}^2 \cdot \mathrm{Tr}\left(\mathbf{A}^{-2}\right)}{T} + \frac{\left\|\mathbf{A}^{-1}\right\|_{\mathrm{op}}^2 \left\|\mu^\star - \theta_0\right\|^2}{T^2}.$$

*and in the case that $\mathbf{\Sigma} = \sigma \mathbf{I}$, it holds that*

$$\mathbb{E}\left[\left\|\widehat{\mu}_T^{\mathsf{EMA}} - \mu^\star\right\|^2\right] \leq \frac{\eta \sigma^2 \cdot \mathrm{Tr}\left(\mathbf{A}^{-2}\right)}{T} + \frac{\left\|\mathbf{A}^{-1}\right\|_{\mathrm{op}}^2 \left\|\mu^\star - \theta_0\right\|^2}{T^2}$$

*Proof.* Note that if we let $\widetilde{\mathbf{A}} = c\mathbf{I}$ and send $c \uparrow \infty$, then we recover $\widehat{\mu}_T^{\mathsf{EMA}} = \widetilde{\mu}_T^{\mathsf{MLE}}(\widetilde{\mathbf{A}})$. Thus, we may apply Theorem 5 to see that

$$\mathbb{E}\left[\left\|\widehat{\mu}_T^{\mathsf{EMA}} - \mu^\star\right\|^2\right] \leq \frac{\eta \left\|\mathbf{\Sigma}\right\|_{\mathrm{op}}^2 \cdot \mathrm{Tr}\left(\mathbf{A}^{-2}\right)}{T} + \frac{\left\|\mathbf{A}^{-1}\right\|_{\mathrm{op}}^2 \left\|\mu^\star - \theta_0\right\|^2}{T^2}.$$

Both results follow immediately from this bound. $\qquad\square$

We also prove a lower bound for $\widehat{\mu}_T^{\mathsf{EMA}}$ as an estimator of $\mu^\star$.

**Proposition 8.** *Let $(\theta_t)_{0 \leq t \leq T}$ be the solution to the OU process (2) with $\mathbf{A} \in \mathbb{R}^{d \times d}$ symmetric positive definite and $\mathbf{\Sigma} = \sigma \mathbf{I}$. Suppose that for some $0 < c < 1$ it holds that $\lambda_{\max}(\mathbf{A})T \leq c/2$. Then*

$$\mathbb{E}\left[\left\|\widehat{\mu}_T^{\mathsf{EMA}} - \mu^\star\right\|^2\right] \geq (1-c)^2 \left\|\mu^\star - \theta_0\right\|^2.$$

*Proof.* We use Lemma 3 and observe that

$$\mathbb{E}\left[\left\|\widehat{\mu}^{\mathsf{EMA}} - \mu^\star\right\|^2\right] = \frac{\left\|\mathbf{A}^{-1}\left(\mathbf{I} - e^{-\mathbf{A}T}\right)\left(\mu^\star - \theta_0\right)\right\|^2}{T^2} + \mathrm{Var}(\widehat{\mu}_T^{\mathsf{EMA}})$$

$$\geq \frac{\left\|\mathbf{A}^{-1}\left(\mathbf{I} - e^{-\mathbf{A}T}\right)\left(\mu^\star - \theta_0\right)\right\|^2}{T^2}.$$

Note that it holds that

$$\mathbf{I} - T\mathbf{A} + \frac{T^2}{2}\mathbf{A}^2 \succeq e^{-\mathbf{A}T} \succeq \mathbf{I} - \mathbf{A}T$$

and thus

$$\mathbf{I} - \frac{T}{2}\mathbf{A} \preceq \frac{\mathbf{A}^{-1}\left(\mathbf{I} - e^{-\mathbf{A}T}\right)}{T} \preceq \mathbf{I}.$$

In particular

$$\lambda_{\min}\left(\frac{\mathbf{A}^{-1}\left(\mathbf{I} - e^{-\mathbf{A}T}\right)}{T}\right) \geq \min\left(1, \left(1 - T\lambda_{\max}(\mathbf{A})/2\right)^2\right)$$

$\qquad\square$

Note that we could have derived a bound for $\widehat{\mu}_T^{\mathsf{MLE}}$ as a special case of Theorem 5, but it would be less tight than that which we derived above; indeed, the simplicity of the model allowed us to precisely characterize the distribution of $\widehat{\mu}_T^{\mathsf{MLE}}$.

### E.4 Proofs related to OUEMA

In this section we prove the additional results mentioned in Section 3, especially with respect to $\widehat{\mu}^{\mathsf{OUEMA}}$. To begin, we prove that $\widehat{\mu}^{\mathsf{OUEMA}}$ is an unbiased estimator of $\mu^\star$.

**Proposition 9.** *Let* $(\theta_t)_{0 \le t \le T}$ *be the solution to the OU process* (2) *with* $\mathbf{A}, \mathbf{\Sigma} \in \mathbb{R}^{d \times d}$ *symmetric positive definite and let*

$$\bar{\theta}_t = \left(\mathbf{I} - e^{-\mathbf{A}T}\right)^{-1} \left(\theta_t - e^{-\mathbf{A}t}\theta_0\right).$$

*Then it holds for any function* $\alpha_T : [0, T] \to \mathbb{R}^d$ *satisfying* $\int_0^T \alpha_T(t)\, dt = 1$ *that*

$$\widehat{\mu}_T^{\mathsf{OUEMA}} = \int_0^T \alpha_T(t)\bar{\theta}_t\, dt$$

*is an unbiased estimator of* $\mu^\star$, *i.e.,* $\mathbb{E}_{\mu^\star}\left[\widehat{\mu}_T^{\mathsf{OUEMA}}\right] = \mu^\star$.

*Proof.* By the definition of $\bar{\theta}_t$ and (11), we have

$$\mathbb{E}_{\mu^\star}\left[\bar{\theta}_t\right] = \left(\mathbf{I} - e^{-\mathbf{A}t}\right)^{-1} \left(\mathbb{E}_{\mu^\star}\left[\theta_t\right] - e^{-\mathbf{A}t}\theta_0\right) = \mu^\star.$$

Now, using the linearity of expectation, we have that

$$\mathbb{E}_{\mu^\star}\left[\widehat{\mu}^{\mathsf{OUEMA}}\right] = \mathbb{E}_{\mu^\star}\left[\int_0^T \bar{\theta}_t \alpha_T(t) dt\right]$$

$$= \int_0^T \mathbb{E}_{\mu^\star}\left[\bar{\theta}_t\right] \alpha_T(t) dt$$

$$= \mu^\star \int_0^T \alpha_T(t) dt = \mu^\star,$$

by the assumption on $\alpha_T$. $\qquad\square$

We now instantiate $\alpha_T$ as a flat average over $[\tau, T]$ for some positive $\tau < T$ and control the variance of $\widehat{\mu}_T^{\mathsf{OUEMA}}$.

**Proposition 10.** *Let* $(\theta_t)_{0 \le t \le T}$ *be the solution to the OU process* (2) *with* $\mathbf{A}, \mathbf{\Sigma} \in \mathbb{R}^{d \times d}$ *symmetric positive definite and for* $0 < \tau < T$, *let*

$$\alpha_T(t) = \begin{cases} 0 & t < \tau \\ \frac{1}{T - \tau} & t \ge \tau \end{cases}.$$

*Then it holds that*

$$\mathbb{E}_{\mu^\star}\left[\left\|\widehat{\mu}_T^{\mathsf{OUEMA}} - \mu^\star\right\|^2\right] \le \frac{\eta \left\|\mathbf{\Sigma}\right\|_{\mathrm{op}}^2 \mathrm{Tr}\left(\mathbf{A}^{-2}\right)}{\left(1 - e^{-\lambda_{\min}(\mathbf{A})\tau}\right)^2 \left(1 - \tau/T\right)^2 \cdot T}.$$

*Proof.* By Proposition 9, we have that $\widehat{\mu}_T^{\mathsf{OUEMA}}$ is an unbiased estimator of $\mu^\star$ and thus the expected squared error is exactly equal to the variance. We can now apply Lemma 3 to see that

$$\mathrm{Var}\left(\widehat{\mu}_T^{\mathsf{OUEMA}}\right) = \mathrm{Var}\left(\frac{1}{T - \tau}\int_\tau^T \left(\mathbf{I} - e^{-\mathbf{A}t}\right)^{-1}\theta_t\, dt\right)$$

$$= \left(\frac{T}{T - \tau}\right)^2 \cdot \mathrm{Var}\left(\frac{1}{T}\int_\tau^T \left(\mathbf{I} - e^{-\mathbf{A}t}\right)^{-1}\theta_t\, dt\right)$$

$$\le \left(\frac{T}{T - \tau}\right)^2 \left\|\left(\mathbf{I} - e^{-\mathbf{A}\tau}\right)^{-1}\right\|_{\mathrm{op}}^2 \cdot \mathrm{Var}\left(\frac{1}{T}\int_0^T \theta_t\, dt\right)$$

$$\le \frac{\eta \left\|\mathbf{\Sigma}\right\|_{\mathrm{op}}^2 \mathrm{Tr}\left(\mathbf{A}^{-2}\right)}{\left(1 - e^{-\lambda_{\min}(\mathbf{A})\tau}\right)^2 \left(1 - \tau/T\right)^2 \cdot T}.$$

The result follows. $\qquad\square$

While we consider the flat average function as a choice for $\alpha_T$ in Proposition 9, the optimal choice of $\alpha_T$ is a different function. Indeed, applying the calculus of variations (Van Brunt, 2004), it is easy to see that the optimal choice of $\alpha_T$ is given (assuming sufficient regularity and finiteness of all quantities) by a scaled version of $\bar{K}_T^{-1} \cdot 1$, where in the case that $\mathbf{\Sigma} = \sigma \mathbf{I}$,

$$
\mathrm{Cov}\left(\bar{\theta}_s, \bar{\theta}_t\right) = \bar{K}_T(s,t) = \frac{\eta \sigma^2}{2}\left(\mathbf{I} - e^{-\mathbf{A}s}\right)\left(\mathbf{I} - e^{-\mathbf{A}t}\right)\mathbf{A}^{-1}\left(e^{-\mathbf{A}|t-s|} - e^{-\mathbf{A}(t+s)}\right),
$$

the equality holds by Lemma 1, the inverse is defined by considering $\bar{K}_T$ as an integral operator, and 1 represents the constant one function. Due to the difficulty of computing the inverse of $\bar{K}_T$, and the fact that we anyhow consider an exponential moving average in practice, we do not pursue this further here.

