# OpenReview forum: "EMA Without the Lag: Bias-Corrected Iterate Averaging Schemes"
_ICLR.cc/2026/Conference — Submitted to ICLR 2026_

### Official Review · Reviewer_YVHS · 2025-10-27

**Soundness:** 2
**Presentation:** 3
**Contribution:** 3
**Rating:** 4
**Confidence:** 3

**Summary:**

This paper proposes BEMA, a new approach to estimate the optimal solution $\mu\^\star$ using the historical trajectory. Theoretically, this proposed method models the optimization dynamic as an OU process and derive the MLE based on this model. The upper bound show that the proposed MLE and OUEMA will converge to the optimal parameter $\mu\^\star$. On the empirical side, this paper proposes a practical approach to obtain the derived estimator and validate its performance in multiple LLM benchmarks. The results show nearly consistent improvements compared to baseline approaches.

**Strengths:**

This paper is well-written and clearly presents its main contributions including a new design to replace EMA in the optimization algorithms.

On the theoretical side, the convergent results seem to be new, as existing aysmptotic results in MLE are mostly based on the sample size instead of the training time T.

The experimental results are significant. The performance gains are visiable in all figures.

**Weaknesses:**

1. The theoretical framework is based on the noisy quadratic model. It is hard to say if this assumption is too strong as all models are quadratic near $\mu^*$ in a sufficiently small neighborhood by using the Tayler expansion. But it indeed restrict the scoop of this theory.

2. It seems that the BEMA requires a few more hyper-parameters, which may not be a desired property for a new approach.

3.  The convergent results built in Section 3 mainly require $\hat{\mu}$ is unbiased; when the matrix A is replaced with an estimator, it seems that the new algorithm doesn't have any convergence guarantees.

4. The experimental results are solid and cover every piece of this new approach. But the experiments are not very standard; it doesn't use any popular benchmark (e.g. nanoGPT speedrun) , instead it chooses their own benchmark.

**Questions:**

Based on the weaknesses, I would have a few questions:

1. As the matrix A may not be a diagonal matrix, it doesn't make too much sense for me to approximate it using simply a diagonal matrix. It is surprising that it "suffices to provide good performance in practice". Is this statement (on page 7, line 350) supported by some experiments or theoretical results?

2. In Adam or AdamW, they are typically combined with an additional second-order momentum term. Are these second-order momentum also included in the BEMA implementation for the comparison experiments (Figure 4)? It can be better if it also includes the comparison with some common optimizer e.g. Adam.

3. How is this proposed approach be scaled in larger models? This paper only considers a few small LLM (~1B), but in modern ML, people will use significantly larger model. It could be more convincing to include some modern architectures with 7B or even larger sizes.

---

> ### Author Response · Authors · 2025-11-12
> **Response to Review**
>
> We thank the reviewer for their careful attention to our work and seek to provide some clarification here.
>
> # SGD vs Adam/AdamW
> Thank you for pointing this out and we will be sure to make this much more clear in the camera ready. The key point is that while our theory is restricted to the SGD setting, **all of our language model experiments are conducted with the Adam optimizer**. For the sake of space we deferred Table 1 to the appendix (p. 19), but will emphasize this important fact in the body. Thus our positive results on BEMA do indeed hold in the standard setting and BEMA demonstrably improves on both using Adam alone and applying EMA to a training trajectory generated by Adam. We apologize for any confusion caused and are happy to provide further clarification as well as interested to know if this changes your opinion on our empirical results.
>
> # Noisy Quadratic model
> We agree that the noisy quadratic model is a simple, analytically tractable regime that does not perfectly capture the complexities of deep network training. Prior work cited in our paper has analyzed the extent to which insights from this model carry over to deep learning optimization and many highly successful algorithmic interventions into optimization, including Adam/AdamW, SOAP, MuON, and even EMA are motivated by analysis in simple linear, quadratic, or convex regimes. To be clear, while our theoretical analysis is used to derive and justify our proposed BEMA, we do not claim that they perfectly capture neural network training dynamics. Our empirical results suggest, however, that even though BEMA was derived in this simpler model, it is still practically quite effective in the deep learning settings we care about.
>
> # Hyperparameter Tuning
> We agree that additional hyperparameters are not ideal, but our ablations (cf. Appendices A and D) suggest that only $\eta$ (the strength of debiasing) has a big effect.  Further investigation to better understand empirical scaling laws of these parameters is an interesting future direction.
>
> # Estimating **A**
> Please see the paragraph beginning in line 276 for a brief discussion on convergence when **A** is unknown as well as Theorem 5 in Appendix E which provides formal bounds; we deferred the theorem to the appendix for the sake of space, but **the key point is that we still have formal convergence results even if A is unknown and they degrade gracefully with how close our estimate is to the true A**.
>
> We will endeavor to clarify the statement in line 350 in the revision, but all we mean to say is that our experiments all use an isotropic **A** and we still see benefit of BEMA over EMA and vanilla training.  This is somewhat surprising, but we expect further gains if we use a less naive approach, which is an interesting direction for further research.  Note that Adam/AdamW has the same problem in that when using Adam, we are approximating the (nondiagonal) Hessian preconditioner with a diagonal matrix in order to satisfy memory constraints imposed by modern hardware.
>
> # Benchmarks
> We agree that further benchmarks are always better, but we emphasize that our current set of results are only meant to be applied to the post-training regime, where optimization is more regular (cf. Lines 302-307).  We attempted to make this point clear in the abstract by restricting our empirical claims to "more stable and efficient *finetuning*" [emphasis added], but will endeavor to further clarify this point in the revision so as to remove any possibility of confusion.
>
> Extending BEMA to the pre-training regime (e.g. nanogpt speedrun) is an interesting future direction and we suspect it requires more careful attention to the choice of $\theta_0$ as well as better estimates of **A**.
>
> # Medium sized models
> We agree that future work should demonstrate the potential for scaling up our approach, but unfortunately our computational resources are not sufficient for large scale experimentation in this regime. We will attempt to include a small study in the 7B setting, but we do not anticipate major changes; if anything, prior work has suggested that optimization becomes simpler and better behaved with more parameters, likely making the simplified toy model more accurate.

---

### Official Review · Reviewer_Ade7 · 2025-10-28

**Soundness:** 3
**Presentation:** 3
**Contribution:** 2
**Rating:** 6
**Confidence:** 4

**Summary:**

This paper proposes BEMA (Bias-Corrected Exponential Moving Average), a theoretically grounded alternative to the widely used EMA for parameter smoothing in stochastic optimization. The authors identify that the exponential averaging of past parameters leads to systematic delay in convergence. Then they derive a bias-free correction based on the Ornstein–Uhlenbeck (OU) formulation of SGD dynamics. The proposed estimator achieves theoretically minimal MSE, matching the lower bound established in Proposition. They empirically verify their theory on small-to-medium language models (Qwen2.5-1.5B, Gemma3-1B, and Llama3.2-1B) across standard instruction-following benchmarks.

**Strengths:**

1. The paper provides a mathematically analysis of EMA under OU dynamics, formally decomposing the bias–variance tradeoff and offering a closed-form correction term.
2. Results show that BEMA consistently improves stability and early-stage convergence compared to vanilla EMA.

**Weaknesses:**

1. A key limitation of this paper is the lack of comparison with modern adaptive optimizers such as Adam or AdamW. All experiments are conducted under the SGD + EMA setting, which is rarely used in contemporary large language model (LLM) fine-tuning. Since Adam itself maintains exponential moving averages of both the first and second moments of gradients, it already implicitly addresses part of the variance–bias tradeoff that the paper analyzes. Therefore, it remains unclear whether the proposed BEMA still provides additional benefits.
2. The experiments are mainly conducted on relatively small or medium-sized models (Qwen2.5-1.5B, Gemma3-1B, and Llama3.2-1B). While the results are consistent with the theoretical analysis, it remains unclear whether the proposed BEMA retains its advantages when scaling to larger models (e.g., 7B–70B).

**Questions:**

Should the $t$ in the left-hand-side of the first equation between line 188-189 be $t+1$?

---

> ### Author Response · Authors · 2025-11-12
> **Response to Review**
>
> We thank the reviewer for their careful attention to our work and seek to clarify some key points here.
>
> # SGD vs Adam/AdamW
> Thank you for pointing this out and we will be sure to make this much more clear in the camera ready.  The key point is that while our theory is restricted to the SGD setting, **all of our language model experiments are conducted with the Adam optimizer**.   For the sake of space we deferred Table 1 to the appendix (p. 19), but will emphasize this important fact in the body.  Thus our positive results on BEMA do indeed hold in the standard setting and BEMA demonstrably improves on both using Adam alone and applying EMA to a training trajectory generated by Adam. We apologize for any confusion caused and are happy to provide further clarification as well as interested to know if this changes your opinion on our empirical results.
>
> # Medium sized models
> We agree that future work should demonstrate the potential for scaling up our approach, but unfortunately our computational resources are not sufficient for large scale experimentation in this regime. We will attempt to include a small study in the 7B setting, but we do not anticipate major changes; if anything, prior work has suggested that optimization becomes simpler and better behaved with more parameters, likely making the simplified toy model more accurate.
>
> # $t$ vs $t+1$
> Thank you for pointing this.  This is a typo and indeed, the $t$ should be $t+1$; the same holds for the $\theta_t$ in that equation.

---

### Official Review · Reviewer_Rogs · 2025-10-30

**Soundness:** 4
**Presentation:** 3
**Contribution:** 2
**Rating:** 4
**Confidence:** 3

**Summary:**

This paper introduces BEMA, a modification of EMA designed to retain its variance-reduction benefits while eliminating lag bias, starting from a stochastic-optimization viewpoint, modeling training trajectories as an OU process, deriving a MLE problem for the target parameter. Autors then propose a practical, drop-in implementation, experimental results demonstrate improved stability and convergence in LMs fine-tuning.

**Strengths:**

Strengths:
* The theoritical results are clearly presented and provides a strong foundation for motivating the proposed method.
* The proposed method is simple to implement and achieves better performance compared with mutiple baselines on accuracy and convergence speed.

**Weaknesses:**

Weaknesses & Questions:

* The motivation of the method is somewhat confusing. The paper begins by stating that closed-loop training requires stabilizers and then proposes BEMA. However, in the supervised fine-tuning setup that this paper focuses on, the training process is not a closed loop scenario. In SFT, the model is trained with a teacher-forcing mechanism, where during the forward pass the next token is predicted based on the ground-truth of the previous token. This differs from autoregressive generation or the setting used in the GVA paper.
* The proposed method introduces additional memory overhead equivalent to the size of the model itself, which could become significant when scaling to larger models.
* The importance of choosing $\theta_0$ as the anchor point remains unclear. It would be valuable to test the sensitivity of the method to anchor selection or provide a discussion on this design choice.
* A recent work [1] also uses $\theta_0$ as an anchor point to accelerate training, which appears conceptually related to this paper. Could you provide a brief discussion with the difference.
* Since the paper emphasizes practical implementation, it would be beneficial to include comparisons regarding resource efficiency, such as memory usage and GPU hours.
* All fine-tuning experiments are conducted on relatively small models (≤ 1.5B) and limited datasets. It remains uncertain whether the same improvements would hold for larger models (e.g., 7B or beyond) or in different training settings, such as instruction tuning.

[1] Harmony in divergence: Towards fast, accurate, and memory-efficient zeroth-order llm fine-tuning

**Questions:**

Some notation is conflict ($\eta$ used twice in line 142 and method part).

Please see weaknesses for questions.

---

> ### Author Response · Authors · 2025-11-12
> **Response to Review**
>
> We thank the reviewer for their careful attention to our paper and seek to clarify several points here.
>
> # Motivation and Closed-Loop Training
> Thank you for pointing this out; we will endeavor to better clarify this point in the revision.  We were motivated by the GVA paper to better understand how we can stabilize training in a way that manifests in improved downstream performance, but we are not claiming to directly address in theory the problem posed by GVA.  As you point out, our theory pertains to a regime where we care about getting as close as possible to the minimizer of the optimization trajectory we observe; this is incidentally helpful for GVA in the sense that that paper posits that the cause of GVA is a much smaller basin of "acceptable" weights $\theta$ when measured by autoregressive rollouts than by the supervised learning loss.  Note that our  *empirical* setting is the same as that of GVA: we are training on teacher-forced data, but care about autoregressive rollouts on LM tasks of interest.  We are happy to provide further clarification if necessary.
>
> # Memory Overhead
> We agree that this could in theory be a problem, but we believe this is addressed in two ways.  First, we can use CPU offloading for the additional copy of the model because we only update BEMA once every large number of steps so the communication and updating time is negligible relative to the rest of training.  Second, in larger scale training, most memory is utilized by passing large batch sizes through the model and so a single additional copy of the model is negligible relative to this cost.  We will add in a discussion of these points in the camera ready.
>
> # Choosing $\theta_0$
> Due to space, we were forced to defer many of our ablations to the appendix.  Please see Appendices A and D for more details.  For this particular question, see Figure 9, where we compare setting $\theta_0$ as the pretrained model as well as the SFT model after 500 and 1K steps.  We agree that further investigation of this choice would be an interesting future direction.
>
> Thank you for pointing out [1]; we will include a discussion in the camera ready.  Briefly, [1] is using 0th order optimization to change the finetuning process itself, whereas we are taking the optimization trajectory as a given and attempting to improve the returned value by treating stabilization as a parameter estimation problem.
>
> # Practical Implementation
> Thank you for pointing this out.  We will endeavor to add this to the camera ready.  Note, however, that the intervention is very similar to the commonly used EMA and thus does not lead to major computational differences relative to this.
>
> We agree that future work should demonstrate the potential for scaling up our approach, but unfortunately our computational resources are not sufficient for large scale experimentation in this regime.  We will attempt to include a small study in the 7B setting, but we do not anticipate major changes; if anything, prior work has suggested that optimization becomes simpler and better behaved with more parameters, likely making the simplified toy model more accurate.

---

### Official Review · Reviewer_SfXw · 2025-10-30

**Soundness:** 3
**Presentation:** 2
**Contribution:** 2
**Rating:** 6
**Confidence:** 4

**Summary:**

This paper proposes Bias-Corrected Exponential Moving Average (BEMA), a new method for stabilizing the training of large language models (LLMs) during fine-tuning. BEMA is a simple and practical augmentation of EMA that eliminates this bias while retaining the variance-reduction benefits. BEMA is a simple extension, requiring only a two-line change to existing EMA implementations. Extensive experiments on LLMs demonstrate that BEMA significantly improves convergence rates and final performance . BEMA's performance is shown to be robust across various optimizer hyperparameters and models.

**Strengths:**

Below are some strengths I find in this paper:
- The authors identify an important issue (the "lag" problem) in traditional EMA when fine-tuning large language models (LMs) with small batch sizes, I think this is a relevant and practical concern in deep learning - resolving which can have a big impact on the quality of downstream applications resulting out of fine-tuning models.
- Stronger theoretical grounding (based on Ornstein-Uhlenbeck process in quadratic optimization) that shows provable acceleration. I think this provides more confidence to me on this method.
- Easy of use: Being a drop-in replacement makes it easy for practitioners to use thie method.
- The level of empirical study conducted to validate this method on real world LLMs models (Gemini, LLama etc) seem promising and assuring to me.

**Weaknesses:**

Despite the strengths, below are some comments I have on areas of imprvoements and some limitations I see:
- Despite the strong theoretical backing using the OU process, the main assumption of having a noisy quadratic model is bit too simplistic in my opinion. It is not clear to me whether the findings nicely carry over to the more complex landscape observed while training real-world Deep Neural Networks and LLMs.
- Not clear how much hyper-parameter tuning went into making this method work. If the authors could add a section or clarify this in much more detail that would be helpful.


Language, Grammar and Typos:
Line 51: The most empirically successful approach to stabilizization is iterate averaging, -> “stabilization”

Use of the term “stabilizer” in the paper is not clear to me. I feel it needs to be explained well, since this is not a standard term in my opinion.

**Questions:**

Some additional qns to the authors:
- I am curious to know what are some challenges authors foresee in adaptively estimating A?
- Authors mention: “observe that training with a fixed learning rate and then applying BEMA leads to the best performance throughout, providing preliminary evidence that applying post-hoc stabilization can obviate the need for learning rate decay in post-training.” -> Is this indeed true? How did the authors confirm this?

---

> ### Author Response · Authors · 2025-11-12
> **Response to Review**
>
> We thank the reviewer for their careful attention to our work.  We now clarify some key points as well as address the questions asked.
>
> # Noisy Quadratic Model
> We agree that the noisy quadratic model is a simple, analytically tractable regime that does not perfectly capture the complexities of deep network training.  Prior work cited in our paper has analyzed the extent to which insights from this model carry over to deep learning optimization and many highly successful algorithmic interventions into optimization, including Adam/AdamW, SOAP, MuON, and even EMA are motivated by analysis in simple linear, quadratic, or convex regimes.  To be clear, while our theoretical analysis is used to derive and justify our proposed BEMA, we do not claim that they perfectly capture neural network training dynamics.  Our empirical results suggest, however, that even though BEMA was derived in this simpler model, it is still practically quite effective in the deep learning settings we care about.
>
> # Hyperparameter Tuning
> For the sake of space, we deferred our many ablations to the appendix (in particular Appendix A for a summary and Appendix D for the results).  You may check for yourself there, but we overall find that BEMA is generally effective at accelerating optimization relative to EMA across a wide array of hyperparameters.
>
> # Use of the term stabilizer
> Thank you for pointing this out; we will endeavor to better explain in the revision.  The main idea is that we treat a stabilizer as any estimator of the minimizer of the optimization process giving rise to the training trajectory.  This term is borrowed from the *Butterfly Effects of SGD Noise* paper but we agree it is not standard and could use further clarification.
>
> # Adaptively Estimating A
> The primary challenge we foresee is in incorporating the adaptive estimation in a memory-efficient and stable way.  The naive approach would be to just use Adam's second moment buffer, but we would need to conduct extensive experiments to understand the empirical scaling of how precisely to do this: e.g., should we actually take an average and scale it by the number of steps or should we keep a running sum of the quadratic variation as our estimate? Empirically, what power should we raise our estimate to and how robust is this choice? Are there ways to incorporate non-diagonal preconditioning that allow for greater acceleration while remaining memory efficient?  These are the first questions we would ask.
>
> # Fixed Learning Rate
> Again, for the sake of space, we were forced to defer our many ablations to the appendix; we will expand on these ablations in the camera ready version.  For this one in particular, please see Figures 6,7,8 and the second bolded paragraph of Appendix D.

---

### Meta-Review · Area_Chair_86HW · 2026-01-05

**Summary:**

This paper introduces Bias-Corrected Exponential Moving Average (BEMA) to stabilize language model fine-tuning. The paper argue that standard EMA reduces gradient variance but introduces a lag that slows optimization. By modeling training dynamics as an OU process, they derive BEMA as a Maximum Likelihood Estimator to correct this bias. The method is presented as a memory-efficient, drop-in replacement for standard EMA. Experiments on models up to 1.5B parameters demonstrate that BEMA accelerates convergence and improves performance on reasoning benchmarks compared to vanilla EMA.

The reviewers cited the paper's rigorous theoretical grounding as primary strengths. They also appreciated the empirical acceleration and improved stability observed on the tested small models. However, several major weaknesses were identified regarding the scale of validation and simplicity of the model. Reviewers criticized the limitation of experiments to models under 2B parameters, arguing this does not prove scalability for modern LLMs. They also noted a disconnect between the theory, which assumes a quadratic landscape and SGD, and the experiments, which utilized Adam. Finally, concerns were raised about the extra memory overhead and the non-standard evaluation benchmarks. All in all, it seems the paper can improve a lot by providing further empirical validation to demonstrate the method's practical utility for large-scale LLMs. The exclusive reliance on small models fails to address whether the benefits persist in regimes relevant to the community. Additionally, the disconnect between the studied model and the experiments should be further discussed and clarified.

**Reviewer Concerns:**

Addressed:
- The authors addressed Reviewer Ade7 and Reviewer YVHS confusion about the choice of the optimizer in the experiments.
- The authors  resolved the concern of Reviewer Rogs about the disconnect between teacher-forced training (SFT) and the closed-loop stability motivation.
- Reviewer YVHS' concern about the choice and knowledge of matrix A is addressed.
- The concerns about hyper parameter sensitivity seems to be reasonably addressed as well.

Still outstanding:
- Model scalability. This was the concern of Reviewers Rogs, Ade7, and YVHS.
- The concerns of Reviewer Rogs about memory overhead is not fully addressed. While the authors suggest CPU offloading, they did not provide any empirical evidence. This concern is important since this is one of the major bottlenecks in fine-tuning LLMs, particularly by smaller/academic labs.
- The concern of Reviewer Reviewer YVHS regarding having non-standard benchmark still holds.

**Reviewer Scores:**

Based on the previous comment, this is my guess:

- Reviewer Ade7 would probably increase their score (since they had mistakes in understanding the experimental setup and the authors' response resolve it.)

- Reviewer YVHS  would probably increase their score  (same reason as the previous reviewer).

- Reviewer Rogs and SfXw would probably maintain their score since their most concerns are not yet addressed (practicality, lack of large-scale experiments, simplistic assumptions)

---

### Decision · Program_Chairs · 2026-01-26

Reject